# New hypotheses of cell type diversity and novelty from orthology-driven comparative single cell and nuclei transcriptomics in echinoderms

Anne Meyer, Carolyn Ku, William L Hatleberg, Cheryl A Telmer, Veronica Hinman*

Department of Biological Sciences, Carnegie Mellon University, Pittsburgh, United States

**Abstract** Cell types are the building blocks of metazoan biodiversity and offer a powerful perspective for inferring evolutionary phenomena. With the development of single-cell transcriptomic techniques, new definitions of cell types are emerging. This allows a conceptual reassessment of traditional definitions of novel cell types and their evolution. Research in echinoderms, particularly sea star and sea urchin embryos has contributed significantly to understanding the evolution of novel cell types, through the examination of skeletogenic mesenchyme and pigment cells, which are found in sea urchin larvae, but not sea star larvae. This paper outlines the development of a gene expression atlas for the bat sea star, *Patiria miniata*, using single nuclear RNA sequencing (snRNA-seq) of embryonic stages. The atlas revealed 23 cell clusters covering all expected cell types from the endoderm, mesoderm, and ectoderm germ layers. In particular, four distinct neural clusters, an immune-like cluster, and distinct right and left coelom clusters were revealed as distinct cell states. A comparison with *Strongylocentrotus purpuratus* embryo single-cell transcriptomes was performed using 1:1 orthologs to anchor and then compare gene expression patterns. The equivalent of *S. purpuratus* piwil3+ Cells were not detected in *P. miniata*, while the Left Coelom of *P. miniata* has no equivalent cell cluster in *S. purpuratus*. These differences may reflect changes in developmental timing between these species. While considered novel morphologically, the Pigment Cells of *S. purpuratus* map to clusters containing Immune-like Mesenchyme and Neural cells of *P. miniata*, while the Skeletogenic Mesenchyme of *S. purpuratus* are revealed as orthologous to the Right Coelom cluster of *P. miniata*. These results suggest a new interpretation of the evolution of these well-studied cell types and a reflection on the definition of novel cell types.

**\*For correspondence:**
vhinman@andrew.cmu.edu

**Competing interest:** The authors declare that no competing interests exist.

## Editor's evaluation

This important study combines single cell and single nucleus transcriptomics to two echinoderm embryos to identify embryonic cell types and assess cell type evolutionary novelties. The paper provides convincing pieces of evidence in regards to the sea star embryonic cell types and their relationship to the sea urchin ones, highlighting conserved, diverged as well as novel genetic programs operating during early echinoderm development. The work will be of broad interest to developmental and evolutionary biologists.

## Introduction

Cell types are the fundamental units of multicellular life. Understanding how novel cell types emerge is critical to understanding the relationships between cells, and hence the processes by which organisms

**Figure 1.** A phylogeny of the major classes of Echinodermata, and Hemichordata outgroup and diagrams of adult and larval phenotypes. Larval schematics show biomineralized larval skeletons colored in red, coelomic pouches colored green, and pigment cells colored in blue. Presence/absence is also indicated using colored circles to the left of the embryos. The pink shading indicates the secretion of biomineralized spicules, but not the formation of an extensive skeleton.

acquire new evolutionary traits. Although fundamental to theories of morphological evolution, understanding how novel cell types evolve, or even how to define a novel cell type, is still unclear and a source of active debate (*Arendt et al., 2016*; *Kin, 2015*; *Marquez zacarias et al., 2021*). Traditionally, a combination of morphology, function, cell lineage, and subsets of gene expression has been used to describe and classify cell types. However, there are many ways in which this method of classification may not accurately reflect evolutionary relationships. For example, cell types with similar morphologies, such as the striated myocytes in bilaterians and cnidarians, appear to be homologous morphologically, when in fact this phenotype arose independently (*Brunet et al., 2016*). Developmental lineage too has drawbacks, as cell types that are very similar in regulatory profiles and morphology may arise from different cell lineages in an embryo. For example, neurons can be derived from both the ectoderm and foregut endoderm in sea urchins (*Angerer et al., 2011*).

Gene Regulatory Networks (GRNs) are a valuable tool for understanding the evolution of cell types within and between species, as evolutionarily related cell types are assumed to have similar regulatory relationships. Homologous cell types can therefore be identified between species by examining the expression of orthologous genes as a proxy for GRNs operating in the cells. However, the cooption of modular GRN subcircuits may frequently occur during evolution, obscuring the relationship between cells in an organism (*Monteiro and Podlaha, 2009*; *Davidson and Erwin, 2010*).

The phylum Echinodermata, which includes species of sea stars, sea urchins, brittle stars, sea cucumbers, and feather stars, is a powerful system for studying the evolution of cell types. Their larval forms in particular have several cell types that, based on developmental, morphological, and functional criteria, are considered to be novel (*Figure 1*). One of the best-studied examples of novel

cell types in these species is the biomineral forming skeletal cells of sea urchin and brittle star (Ophiuroidea) pluteus larvae. These cells derive from early embryonic skeletogenic mesenchyme that ingress and secrete a calcium-based biomineral skeleton within the blastocoel. The long skeletal rods formed by this process give the larvae the distinctive pyramidal, plutei shape (*Decker and Lennarz, 1988*). The larval forms of the other, non pluteus forming, groups of echinoderms—the bipinnaria in sea stars, auricularia in sea cucumbers, auricularia-like in crinoids, and the tornaria larva of the outgroup Hemichordate phylum—do not produce larval skeletal rods. The sea cucumber auricularia does however produce a small biomineralized cluster that appears to be secreted from mesenchyme cells (*Decker and Lennarz, 1988*; *Hyman, 1955*; *McCauley et al., 2012*).

Another novel cell type, the larval pigment cells, are unique to the sea urchins (i.e. the echinoid family). The pigment cells emerge in the second wave of mesenchymal ingression during gastrulation and embed themselves in the ectoderm of the larvae (*Gustafson and Wolpert, 1967*; *Massri et al., 2021*). Upon infection, the immune response is activated and these cells ingress into the blastocoel and aggregate with other immune cells at sites of high pathogen concentration (reviewed in *Buckley and Rast, 2017*). The molecule that gives these cells their distinctive color is echinochrome A, a naphthoquinone with known antimicrobial properties (*Service and Wardlaw, 1984*).

Much is known about the mechanisms of specification of these novel cell types in sea urchins, especially the purple sea urchin, *Strongylocentrotus purpuratus,* and the green sea urchin, *Lytechinus variegatus*, which have an especially well-resolved early developmental GRN (*Davidson et al., 2002a*; *Davidson et al., 2002b*; *Hinman et al., 2003a*; *Oliveri et al., 2002*; *Saunders and McClay, 2014*; *Smith et al., 2018*). These GRNs have provided the basis for comparison with several other echinoderm taxa, most extensively the bat sea star, *Patiria miniata* (*Cary et al., 2020*; *Hinman and Cheatle Jarvela, 2014*; *Hinman et al., 2009*). Sea stars such as *P. miniata* morphologically lack both larval-skeleton-forming cells and pigment cells (*Figure 1*).

Comparisons of GRNs within different cell types have been used to infer the evolution of the sea urchin skeleton and pigment cell types from an ancestor shared with sea stars. A leading hypothesis from these studies is that the sea urchin and brittle star larval skeleton arose through the co-option of the adult skeletogenesis program by a subpopulation of embryonic mesodermal cells (*Gao and Davidson, 2008*; *Morino et al., 2012*), under the regulatory control of the transcription factor *aristaless-like homeobox* (*alx1*) and cooption of the *vascular endothelial growth factor receptor (vegfr)* (*Ben-Tabou de-Leon, 2022*; *Ettensohn and Adomako-Ankomah, 2019*; *Morino et al., 2012*). These regulatory genes are required for the expression of genes needed for biomineralization in these sea urchin cells (*Duloquin et al., 2007*; *Rafiq et al., 2012*). *vegfr* in particular had not been detected in echinoderm embryos that do not form skeletons. Likewise, the pigment cells are proposed to have arisen through a heterochronic co-option of an adult program for the formation of these cells (*Perillo et al., 2020*).

Single-cell RNA-sequencing (scRNA-seq) has revolutionized our understanding of cell-type diversity within an organism (*Konstantinides et al., 2018*; *Marioni and Arendt, 2017*; *Morris, 2019*; *Shapiro et al., 2013*; *Tanay and Sebé-Pedrós, 2021*). This technology profiles the transcriptome of individual cells within an organism, preserving the heterogeneity of cell states that are lost in traditional bulk sequencing. This new technology provides prominence to gene expression and gene regulatory networks (GRNs) as the basis of cell-type identity and the evolutionary relationships between cell types (*Achim and Arendt, 2014*; *Arendt, 2008*; *Arendt et al., 2016*). Knowledge of echinoderm cell diversity and molecular profiles has recently benefited greatly from scRNA-seq studies in echinoids (*Foster et al., 2020*; *Massri et al., 2021*; *Paganos et al., 2021*). These datasets provide an unbiased view of the gene expression state of cell types in sea urchin embryos. This presents an opportunity to establish equivalent profiling in sea stars to establish the similarities and differences of gene expression states on a large scale rather than the candidate gene approaches that have been performed to date. We, therefore, developed a single-nucleus RNA-seq (snRNA-seq) developmental cell atlas for *P. miniata*. snRNA-seq is performed on isolated nuclei rather than whole cells (*Lake et al., 2016*) and is suggested by recent literature to provide a more representative view of cell types than single-cell sequencing because the uniform nuclear morphology permits less biased isolation (*Wu et al., 2019*).

To compare expression states across species, we made a multi-species developmental atlas from previously published *S. purpuratus* scRNA-seq data (*Foster et al., 2020*) using an integrated approach based on the expression patterns of a comprehensive mapping of 1:1 gene orthologs between *S.*

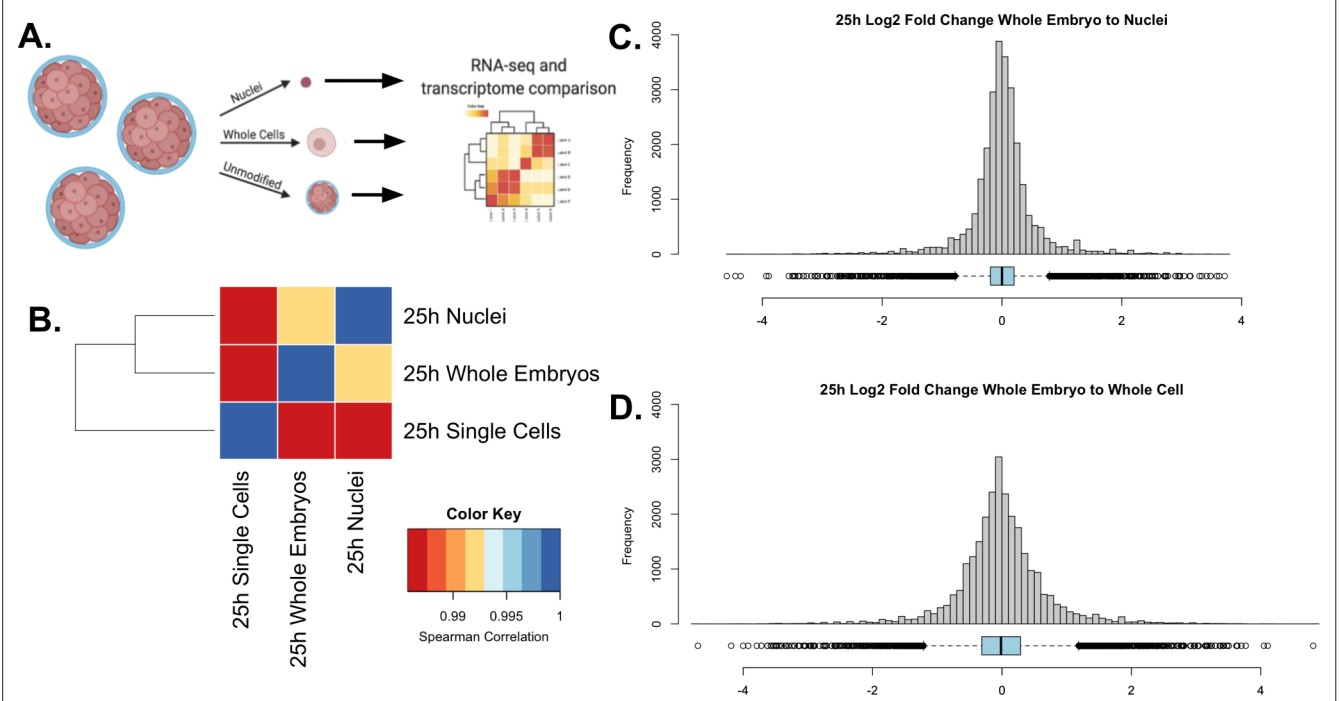

**Figure 2.** Validation of single nuclear RNA-seq protocol. (**A**) A schematic representation of the experimental design. A single culture of 25 hpf *P. miniata* embryos was partitioned into three aliquots: 1. Whole embryos, 2. Nuclei, and 3. Single cells. RNA was extracted and sequenced and transcriptomes were compared across samples. (**B**) Spearman correlation was conducted on quantile-normalized gene counts from the RNA-seq dataset. Nuclear isolation had the highest correlation with the whole embryo RNA-seq dataset with $\tau$ =0.991. The whole-embryo to the whole-cell Spearman correlation coefficient of $\tau$ =0.986. Single cells to nuclei has a coefficient of $\tau$ =0.987. (**C & D**) Histograms and dot plots display the frequency distribution of log2 fold changes in gene expression levels between the whole embryo and nuclear or whole-cell data sets. (**C**) The nuclear RNA-seq datasets resulted in a distribution where 5=0.013, 5=0.558, and M=0.000.(**D**) The single-cell to whole embryo comparison resulted in a distribution where 5=−0.003, 5=0.683, and M=−0.013.

The online version of this article includes the following figure supplement(s) for figure 2:

**Figure supplement 1.** Single nucleus atlas of early development in *S. purpuratus.*

*purpuratus* and *P. miniata* (***Arshinoff et al., 2022***; ***Beatman et al., 2021***; ***Foley et al., 2021***). For each gene discussed in this paper, the species' NCBI-curated locus IDs (LOCID) and associated nomenclature established in ***Beatman et al., 2021*** are presented in ***Supplementary file 1***. We use these atlases to compare the GRN definition of cell type novelty to the morphological definition, to establish alternative hypotheses of the origin of pigment cells and skeletogenic mesenchyme and to broadly reflect on the definitions of cell type and novelty.

# Results

## Single-nucleus transcriptome is representative of cell diversity in sea star embryos

Before creating our *P. miniata* atlas, we wanted to explore the use of snRNA-seq in echinoderms to see if it offers any benefit over scRNA-seq, in particular a more robust isolation. We, therefore, adapted protocols for single nuclear isolation used in Omni-ATACseq (***Corces et al., 2017***). This is the first time that a single nuclei sequencing protocol has been conducted in an echinoderm.

We sought to establish whether our nuclear isolation method was less biased in cell-type retention by comparing bulk-RNA seq data collected from samples isolated using whole-cell isolation, nuclear isolation, or from whole embryos (***Figure 2A***). We compared the overall similarity of transcriptome profiles using the Spearman correlation coefficient for each dataset (***Figure 2B***). The nuclei-to-whole embryo comparison had the highest Spearman coefficient (ρ=0.987), indicating higher overall transcriptome similarity to the undissociated embryos. Whole-cell isolation also showed a strong

correlation with the whole embryo, with a Spearman coefficient of ρ=0.985, demonstrating that, like nuclear isolation, single-cell isolation still reflects the overall transcriptomic profile of the whole organism.

To further assess nuclear versus cellular isolation bias, we calculated the log2 fold change in gene transcript count for each of our isolation methods with respect to the intact embryo control (*Figure 2C-D*). The nuclear to whole-embryo comparison resulted in a mean of 0.013 and a standard deviation of 0.558. The whole-cell to whole-embryo comparison resulted in a mean of –0.003 and a standard deviation of 0.683. Although the mean is closer to zero in the whole-cell isolates, we conclude that a nuclear isolation method more consistently reflects whole embryo expression diversity because of the lower standard deviation, indicating the overall distribution is more narrow and therefore more genes accurately reflect the expression levels of a whole organism.

Finally, as proof of principle, we performed snRNA-seq in *S. purpuratus* to determine if we could identify distinct clusters of known cell types. Using samples from four time points [6 hours post-fertilization (hpf), 15 hpf, 23 hpf, and 33 hpf], we were able to identify 15 clusters, encompassing all three germ layers and including several clusters corresponding to skeletogenic mesenchyme and pigment cells. These skeletogenic mesenchyme clusters were characterized by gene expression associated with different degrees of differentiation. Further analysis of this pilot dataset is provided in *Supplementary file 5* and *Figure 2—figure supplement 1*. These results thus strongly support that single nuclei and single-cell RNA isolation and sequencing each resulted in appropriate representative clusters thus providing confidence that either isolation method was an effective way to establish cell clusters. snRNA-seq is more likely to capture earlier transcriptional responses than scRNA-seq, but such timing differences are much smaller than the sampling times here and unlikely to affect our analyses.

## A cell type atlas for *Patiria miniata* embryo development

Previous work has shown that pooling scRNA-seq data from the eight-cell stage to the late gastrula of *S. purpuratus* into one atlas results in biologically meaningful clustering (*Foster et al., 2020*). Therefore, we conducted snRNA-seq at three different time points within an early developmental window spanning from blastula to mid-gastrula stage, to compile an atlas describing early development in *P. miniata*. snRNA-seq was conducted on *P. miniata* embryos at three different time points (16 hpf, 26 hpf, 40 hpf) (*Figure 3B*). These time points span from blastula (16 hpf), early gastrulation (26 hpf) and mid-gastrulation (40 hpf) when cell types of the larva are first specified (*Cary et al., 2020*; *Yankura et al., 2010*).

We isolated nuclei and performed single nuclear sequencing, processed reads, and aligned them to a pre-mRNA index of the recently assembled *P. miniata* genome v3.0 (*Arshinoff et al., 2022*). After data filtration and quality control (see Methods), we were left with 17,188 nuclei. Dimensional reduction and clustering (see Methods) resulted in 23 clusters (abbreviated with Pmin CL) (*Figure 3*). Cluster size ranged from 133 nuclei (Pmin CL 22 'Sensory Neural') to 1739 nuclei (Pmin CL 0 'Mesoderm-like').

All clusters contain nuclei from each of the time points indicating that there are no substantial batch effects (*Figure 3D* and *Figure 3—figure supplement 1*). This also suggests that distinct transcriptional states are also present by 16 hph, much earlier than evident by morphological features. However, there are some time point enrichments found in the clusters. Clusters 7, 18, and 22 are dominated by cells from the later time points, with only 10.6%, 3.27%, and 5.09% cluster members originating from the earliest time point respectively. Clusters 10, 14, and 21 are dominated by cells from the earlier time points, with 63.0%, 49.6%, and 56.1% originating from the 16 hpf dataset. These observations suggest that differences in developmental timing may have an impact on transcriptome profiles and therefore should be kept in mind during cluster annotation.

Representatives from all germ layers were identified through cluster annotation, as is displayed in *Figure 4*. This figure presents a subset of genes, whose expression patterns were used to inform the naming of each cluster. Cluster annotation was performed by calculating genes that are differentially expressed in a cluster relative to the rest of the dataset, henceforth known as marker genes (see methods for marker gene calculation and classification). Additionally, genes with known expression patterns (from previously published literature) and highly conserved cell-type-specific functions (such as enzymes involved in neurotransmitter synthesis) also helped inform cluster annotation. Finally, we

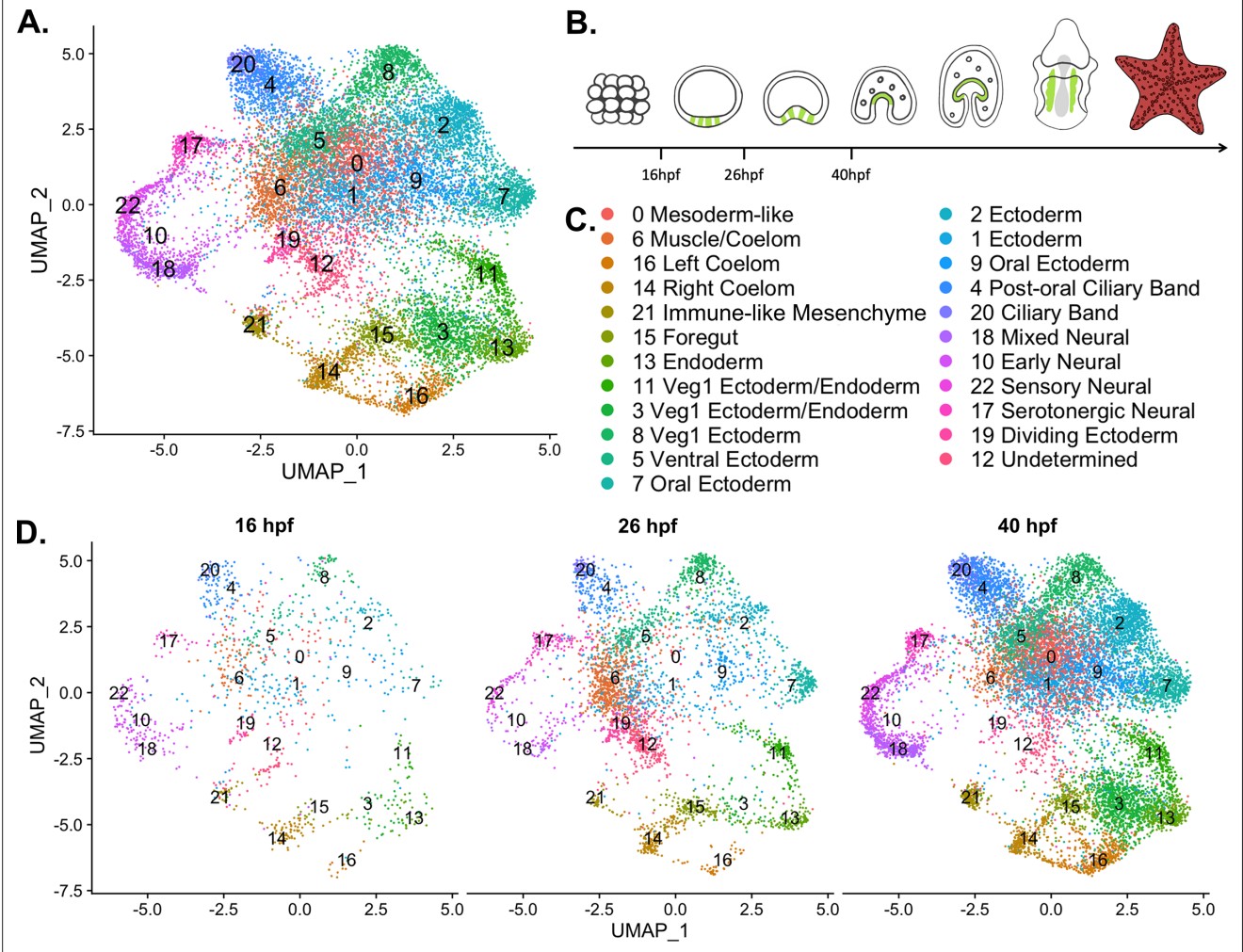

**Figure 3.** A nuclear atlas for early *P. miniata* development. (**A**) The UMAP projection of the *P. miniata* atlas developed from snRNA-seq of three developmental time points. 17,118 nuclei were considered in total. (**B**) The schematic shows cartoons of stages for which time points were sampled. The green color indicates the coelomic pouches. (**C**) A key displaying the cluster names corresponding to UMAP projections in **A and D**. (**D**) The UMAP projection of the *P. miniata* nuclear atlas, separated by the three-time points of sampling, 16 hpf has 1076 nuclei, 26 hpf has 3888 nuclei, and 40 hpf has 12,203 nuclei.

The online version of this article includes the following figure supplement(s) for figure 3:

**Figure supplement 1.** Contribution from each time point to the *P. miniata* developmental atlas clusters.

performed whole-mount in situ hybridization (WMISH, see methods) on a selected group of genes to validate cluster annotations (*Figures 5–7*).

Seven of the 23 identified clusters (Pmin CL 17, 22, 18, 10, 21, 14, and 16), are discussed in detail below and will be referenced in our subsequent comparative analysis. Four of these clusters (Pmin CL 17, 22, 18, 10) represent neuronal cell types, offering valuable insight into the diversity of neurons in early echinoderm embryonic development. One cluster represents a distinct immune-like population (Pmin CL 21), which is the first time this cell type has been molecularly characterized in this species. Finally, we identified two populations of coelomic cells, left and right (Pmin CL 14 and 16), indicating that these tissues are composed of transcriptomically distinct populations. A complete description of the cluster annotation justification for clusters not discussed in this paper's body is provided in *Supplementary file 6*.

## Neural clusters

The four neural clusters (Pmin CL 17, 22, 18, 10) are all closely associated in the UMAP projection (*Figure 5B*). Their identities were established based on the expression of known neural markers in *P.*

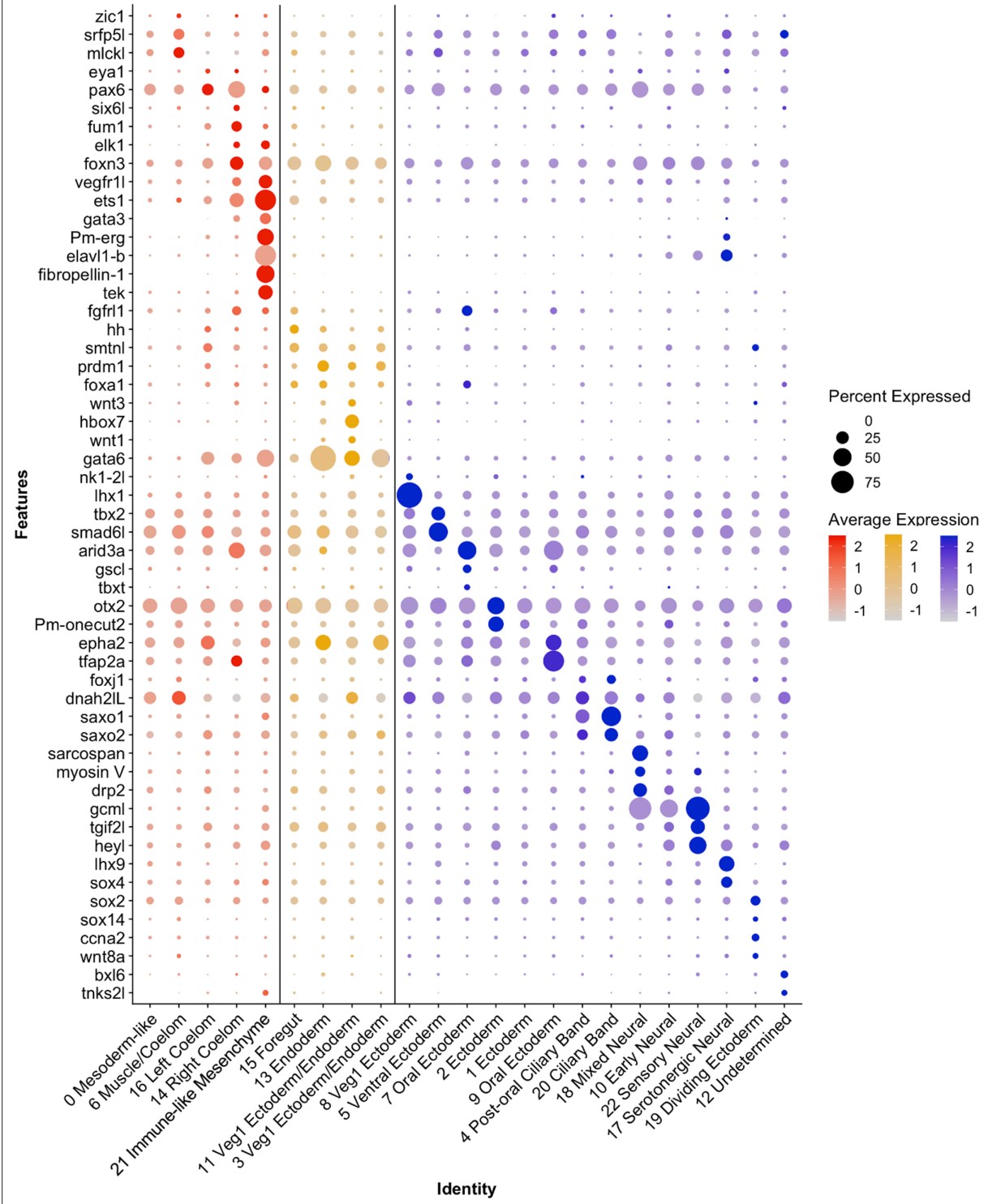

**Figure 4.** Differential expression of *P. miniata* marker genes was used to annotate cluster identity. A dot plot highlighting marker genes used to determine cluster identity. The X-axis lists the cluster names, while the Y-axis lists gene names. Mesodermal clusters are colored in red, endoderm in yellow, and ectoderm/undetermined in blue. Circle size corresponds to the number of cells in the cluster expressing the gene of interest, while shade correlates with the level of expression.

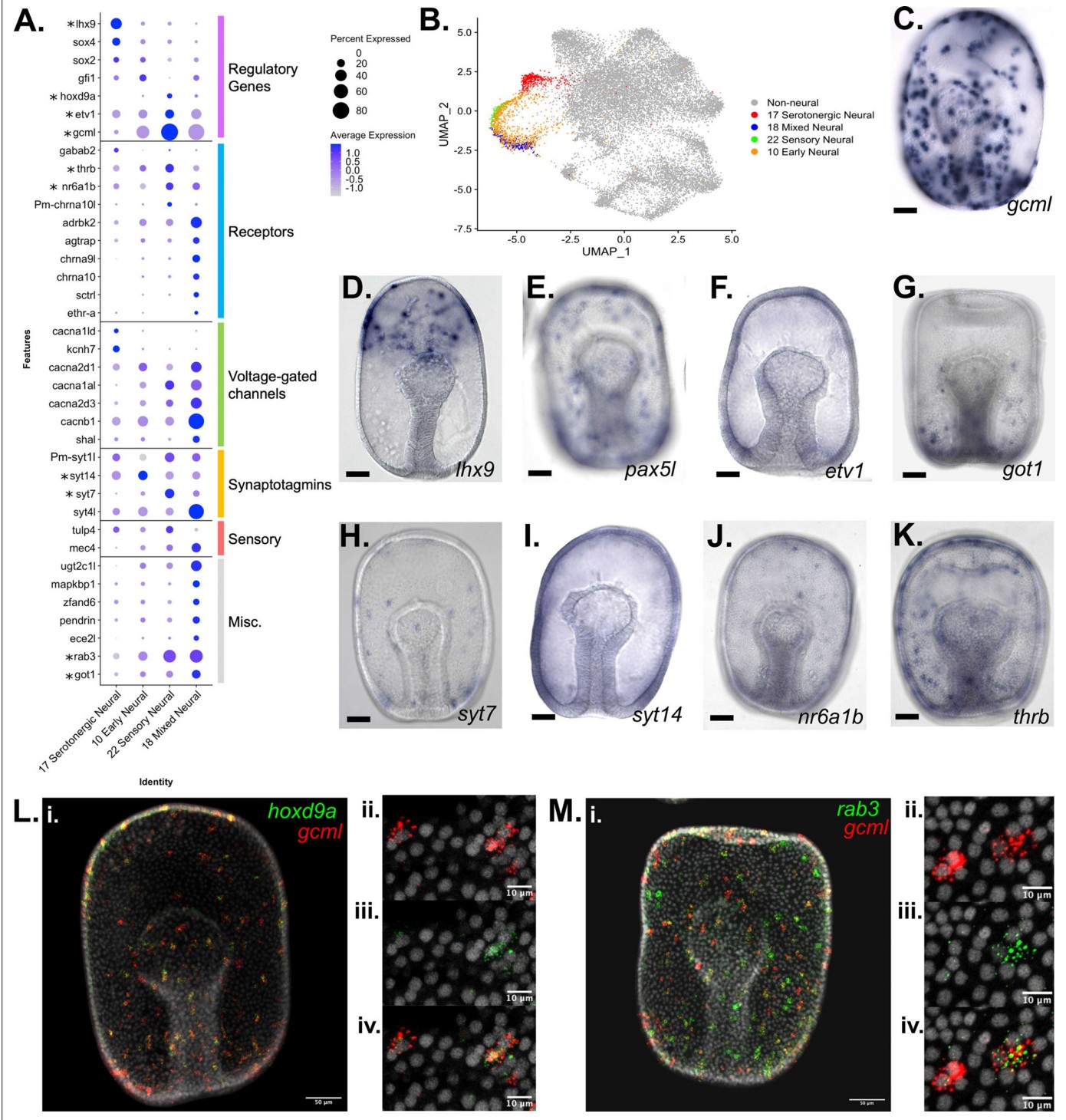

**Figure 5.** Characterization of neuronal populations in the *P. miniata* developmental atlas. (**A**) A dot plot of the four neural clusters and a selection of genes used to annotate the clusters. The genes are divided into six categories: genes encoding: regulatory proteins, receptors, voltage-gated channels, synaptotagmins, sensory, and miscellaneous. Genes with in situs are marked with an asterisk. (**B**) A UMAP projection highlights the four neural clusters shown color coded. (**C–K**) Representative images of WMISH for validation of marker gene expression at the gastrula stage embryo (48 hpf). (**D**) *gcml* is expressed in cells throughout the ectoderm, with an exclusion zone on the oral, non-neurogenic region of the gastrula. (**D**) *lhx9* is expressed in serotonergic neuronal precursors near the animal pole. (**E**) *pax5L* is expressed in cells embedded throughout the ectoderm, with an exclusion zone on the non-neurogenic oral surface (observed 46/46 gastrula). (**F**) *etv1* is expressed in cells embedded throughout the ectoderm (observed in 19/19 gastrula). (**G**) *got1* expression is observed in cells dispersed throughout the ectoderm (observed in 31/31 gastrula). (**H**) *syt7* is expressed in

*Figure 5 continued on next page*

Figure 5 continued

cells embedded in the ectoderm, with the greatest abundance in the central and vegetal ectoderm (observed in 14/15 gastrula). (**I**) *syt14* has broad expression across ectodermal cells (observed in 15/18 gastrula). (**J**) *nr6a1b* is expressed in cells embedded in the ectoderm (observed 38/38 gastrula). (**K**) *thrb* is expressed in cells embedded throughout the ectoderm (observed 32/33 gastrula). (**L–M**) Double FISH examination of additional marker genes (*rab3* and *hoxd9a*) together with *gcml*. (**L.i**) *gcml* is colored red and *hoxd9a*, in green, are expressed in cells scattered throughout the ectoderm (WMISH validates this pattern of *hoxd9a* expression, with ectodermal embedding observed in 20/20 gastrula). Higher magnification microscopy shows that *gcml* and *hoxd9a* are co-expressed in some cells, while others express only *gcml*. (**M**) (**M.i**) *gcml* marked in red and *rab3*, in green, are expressed in cells embedded throughout the ectoderm (WMISH validates this pattern of *rab3* expression, with ectodermal embedding observed in 22/23 gastrula). Higher resolution microscopy (**M.ii-iv**) shows that *rab3* is expressed in a subset of *gcml*-expressing cells. Scale bars on images C-K are 50 μm.

miniata and genes with well-documented, conserved neural function including pre-synaptic synaptotagmins, neurotransmitter receptors, and the presence of voltage-gated ion channels (***Adolfsen and Littleton, 2001***; ***Burke et al., 2006***; ***Moghadam and Jackson, 2013***; ***Walker and Holden-Dye, 1991***; ***Figure 5A***).

The marker gene sets for these four neural clusters include four synaptotagmin genes: *synaptotagmin-7* (*syt7*), *synaptotagmin-14* (*syt14*), *synaptotagmin-1-like* (*syt1l*), and *synaptotagmin-like protein 4* (*syt4l*). The synaptotagmin gene family has been extensively documented to have conserved functionality in neurons across phyla (***Adolfsen and Littleton, 2001***). *syt14* is expressed in all neural clusters but is most highly expressed in Pmin CL 10. WMISH confirms that *syt14* is expressed broadly throughout the ectoderm (***Figure 5I***) in large puncta reminiscent of a neural pattern. Conversely, *syt1l* is expressed in all clusters but shows the lowest expression in Pmin CL 10. *syt4l* is expressed in all clusters, but most strongly in Pmin CL 18 (***Figure 5A***). WMISH shows that *syt7* is expressed in a few cells embedded in the gastrula's ectoderm (***Figure 5H***), as opposed to the more broad expression of *syt14* across the ectoderm (***Figure 5I***), suggesting that synaptotagmin expression is not uniform across different neural populations. This suggests that in echinoderms, members of the synaptotagmin gene family are differentially expressed between different classes of neurons, as they are in *Drosophila* and Mammalia (***Adolfsen et al., 2004***; ***Craxton, 2004***; ***Marqueze et al., 1995***).

In this analysis, we also identified several new markers for neural populations in *P. miniata*. *ETS variant transcription factor 1* (*etv1*) is a transcription factor expressed in all four neural clusters and expressed in cells dispersed throughout the ectoderm (***Figure 5F***). Likewise, *glutamic-oxaloacetic transaminase 1* (*got1*) is found in three of the four clusters and expressed in a manner spatially consistent with neural fate (***Figure 5G***).

## Serotonergic neural

*LIM homeobox 9* (*lhx9*), a gene previously established as a marker of serotonergic neurons (***Jarvela et al., 2016***), was found to be a strong marker of Pmin CL 17 and was verified using WMISH (***Figure 5C***). *Tryptophan 5-hydroxylase 1-like* (*tph1l*), a gene involved in serotonin biosynthesis (***Fitzpatrick, 1999***) has been shown to localize to serotonergic neurons in sea urchins (***Yaguchi and Katow, 2003***), was also identified as a marker of this cluster. This along with the expression of other known markers including transcription factor *SRY-box transcription factor 4* (*sox4*), *neogenin-like* (*neo1l*), and *ELAV-like protein 1B* (*elavl1-b*) allow us to conclude Pmin CL 17 corresponds to serotonergic neurons (***Yankura et al., 2013***).

## *Gcml*-expressing neural cluster

*Glial cells missing transcription factor-like* (*gcml*), the sea star ortholog of the *Drosophila* gene glial cells missing (Dmel_CG12245), was also identified as a statistically significant marker gene of Pmin CL 22 and showed increased expression in Pmin CL 10 and Pmin CL 18, relative to the serotonergic cluster. WMISH verified that this gene localizes to a population of cells embedded throughout the ectoderm (***Figure 5D***). Another gene we see upregulated in these *gcml*-expressing clusters is *rab3 GTPase* (*rab3*). Double FISH (***Figure 5M.i-iv***) of *rab3* and *gcml* shows both genes expressed in cells embedded in the ectoderm. When examining the localization patterns of these genes (***Figure 5M.ii-iv***), we see *rab3* expressed in only a subset of *gcml*-positive cells. This suggests that *gcml* marks a broad category of non-serotonergic neurons, a subset of which also express *rab3*.

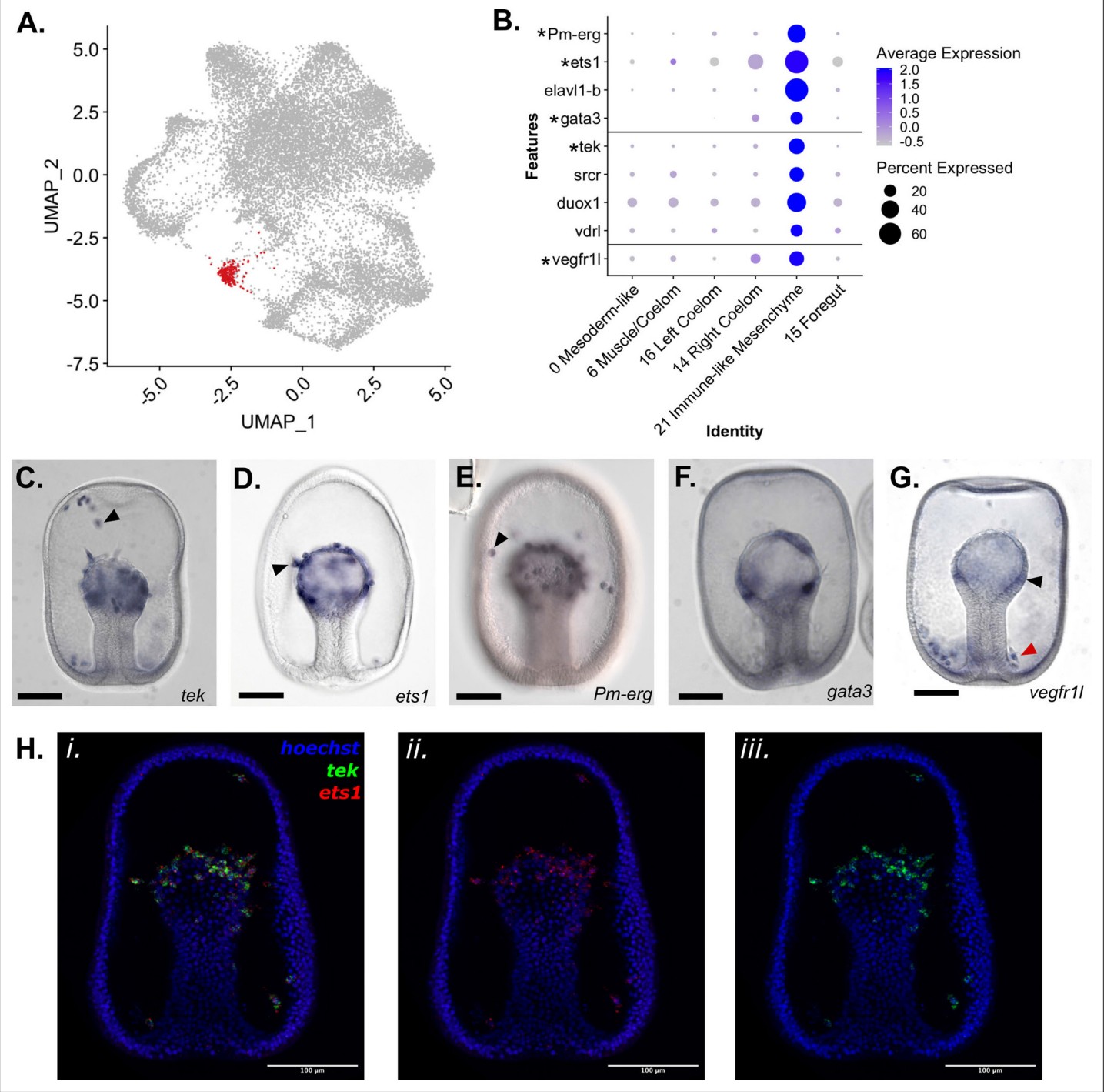

**Figure 6.** Identification of an Immune-like Mesenchyme cell type in *P. miniata*. (**A**) A UMAP projection highlights in red the cluster annotated as Immune-like Mesenchyme. (**B**) A dot plot of mesodermal clusters and the genes used to identify the Immune-like Mesenchyme cluster identity. Genes with in situs are indicated with an asterisk. (**C–G**) Representative images of WMISH for validation of marker gene expression at the gastrula stage embryo (48 hpf). (**C**) *tek* is expressed in the mesenchyme (indicated with an arrow) and mesodermal bulb (observed in 33/34 embryos). (**D**) *Ets1* is expressed in mesodermal cells undergoing the epithelial to mesenchyme transition (indicated with an arrow). (**E**) *Pm-erg* is expressed in mesenchymal cells (indicated with an arrow) as well as in the mesodermal bulb. (**F**) *Gata3* is expressed in patches of the mesodermal bulb (observed 27/30 embryos). *vegfr1l* is expressed in the mesenchyme (indicated with a red arrow) and mesodermal bulb (indicated with a black arrow) (observed 51/55 embryos). This mesenchyme clusters in a ring around the base of the archenteron (localization observed in 28/36 embryos). (**H**) Double FISH shows coexpression of *ets1* (ii.) and *tek* (iii.) in the mesenchyme gastrula. Hoechst is used to label the nuclei. Scale bars on images C-G are 100 µm.

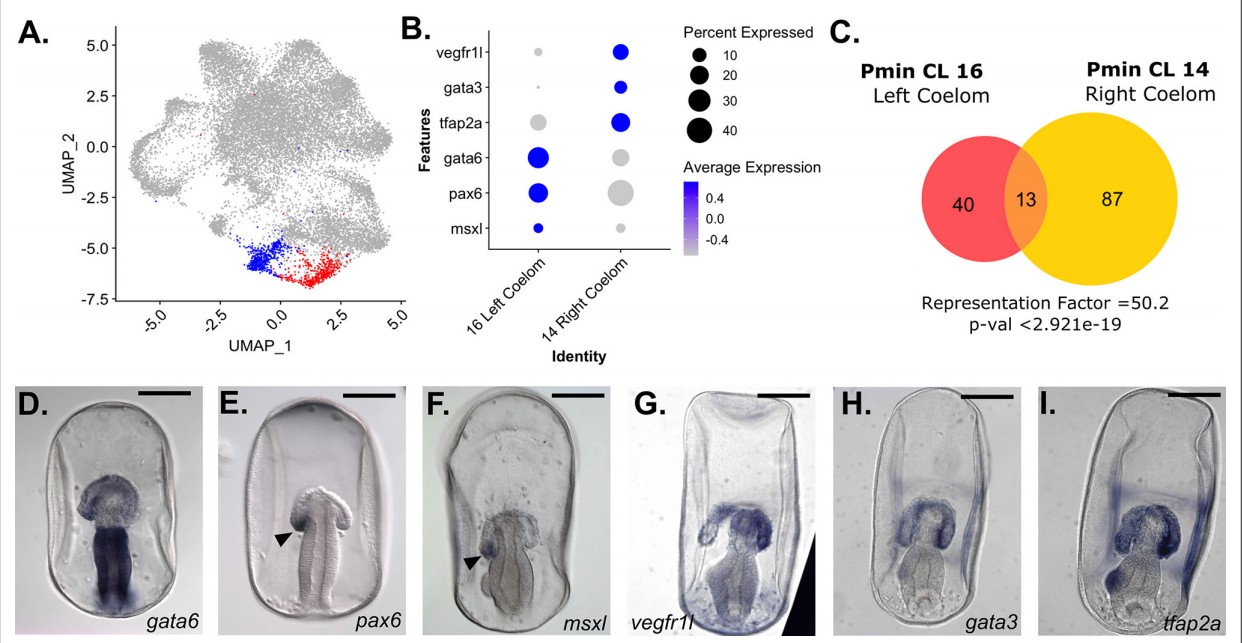

**Figure 7.** snRNA-seq detects left/right asymmetry in *P. miniata* coelom development. (**A**) A UMAP projection of the *P. miniata* atlas where blue highlights the cluster for the right coelomic pouch and red highlights the cluster for the left coelomic pouch. (**B**) A dot plot of two coelom clusters and the genes identified as differentially expressed between the two clusters. (**C**) A Venn diagram shows the overlap in marker genes shared between the two coelomic cells. Thirteen genes are shared in total, with the overlap yielding a representation factor of 50.2 with a p-value <29.21e-19. Representative images of WMISH validation of genes marking the left (**D–F**) and right (**G–I**) in late gastrula stage embryos (72 hpf). *gata6*, *pax6*, and *msxl* are expressed in the left coelomic pouch (indicated with an arrow in **E** and **F**) and absent in the right pouch. *vegfr1l*, *gata3*, and *tfap2a* are expressed in both pouches but are more highly expressed in the right coelomic pouch compared to the left. Scale bars on images D-I are 100 μm.

## Sensory Neural

Pmin CL 22 was annotated as a sensory neuron population. Genes relating to neurotransmitter receptors were also found to be expressed in this cluster. These include neuronal *acetylcholine receptor subunit alpha-10-like (chrna10l)*, and *beta-adrenergic receptor kinase 2 (adrbk2)*. It was also marked by the expression of *thyroid hormone receptor beta (thrb)* and *nuclear receptor subfamily 6 group A member 1B (nr6a1b*, previously known as *steroid hormone receptor 3*). WMISH of these two genes confirmed localization to a subset of ectoderm, consistent with a neural population (*Figure 5F-G*). This cluster was also represented by high levels of the transcription factor *paired box protein Pax5-like (pax5l)* which was also shown via WMISH to express in cells embedded in the ectoderm (*Figure 5E*).

Finally, Pmin CL 22 was marked by genes related to a diverse range of sensory capabilities such as mechanosensing (*degenerin mec-4 (mec4)*) (*O'Hagan et al., 2005*) and photosensing (*TUB like protein 4 (tulp4)*) (*Carrella et al., 2020*).

Pmin CL 22 was also marked by the expression of transcription factor *homeobox protein Hox-D9-like (hoxd9a)*, a gene not previously examined in echinoderms. Double FISH of *hoxd9a* and *gcml* (*Figure 5L.i-iv*) shows both genes expressed in cells embedded in the ectoderm. As with the *rab3-gcml* double in situ (*Figure 5M.i-iv*), we see *hoxd9a* expressed only in a subset of *gcml*-positive cells (*Figure 5L.ii-iv*). This further supports our hypothesis that *gcml* is involved in the specification of several different neural populations, including a population that expresses *hoxd9a* in addition to *gcml*.

## Mixed Neural

Pmin CL 18 corresponded to a population of cells of mixed neural specification. As with Pmin CL 22, this population is distinct in its higher expression of *gcml* and *pax5l*. It also is defined by the expression of neurotransmitter receptors neuronal *acetylcholine receptor subunit alpha-9-like (chrna9l)*, neuronal *acetylcholine receptor subunit alpha-10 (chrna10)* and hormonal receptors including *nr6a1b*, *type-1 angiotensin II receptor-associated protein-like (agtrap)* and *ecdysis triggering hormone receptor subtype-A (ethr-A)*. Several other genes related to neuroendocrine function were also present, such

as *endothelin-converting enzyme 2-like* (*ece2l*) (*Mzhavia et al., 2003*), and *pendrin* (*Royaux et al., 2000*).

Pmin CL 18 is also marked by the expression of *serotonin N-acetyltransferase-like* (*aanat*). Pmin CL 18 is also characterized by the expression of highly conserved muscle-associated genes such as *dystrophin-related protein 2* (*drp2*; *Huang et al., 2004*) and *sarcospan* (*Hooper and Thuma, 2005*; *Lehman and Szent-Györgyi, 1975*).

Genes potentially related to immune function were also detected as markers including *UDP-glucuronosyltransferase 2C1-like* (*ugt2c1l*) (*Wang et al., 2021*), *Scavenger receptor cysteine-rich domain superfamily protein* (*srcr*) (*Pancer et al., 1999*) and several genes linked to the NF5B pathway (*mitogen-activated protein kinase-binding protein 1* (*mapkbp1*) (*Fu et al., 2015*) and *AN1-type zinc finger protein 6* (*zfand6*) *Huang et al., 2004*).

## Early Neural

Finally, Pmin CL 10 was also annotated as a non-specific early neuronal population. This cluster is marked by the expression of *pax5l* and *adrbk2*. Although not a statistically significant marker gene, this cluster also expresses *agtrap*. When we examine the normalized contribution of the three sampling time points to each cluster (*Figure 3—figure supplement 1*), we see that our earliest time point (6 hpf) makes up 63% of cluster 10, as compared to 5% of cluster 22, 3% of cluster 18, and 34% of cluster 17. Therefore, Pmin CL 10 may represent an earlier population of neurons, whereas the other three neural clusters are more differentiated neuronal populations.

## Immune-like Mesenchyme

Pmin CL 21 is a very spatially distinct cluster in the UMAP dimensional reduction plot (*Figure 6A*) and was identified as Immune-like Mesenchyme. We annotated this cluster as mesenchymal in nature based on the expression of known mesenchymal markers, that is *ETS proto-oncogene 1 transcription factor* (*ets1*), *transcriptional regulator ERG homolog* (*Pm-erg*). The nuclei of this cluster also express genes involved in immune response in other echinoderms. *TEK receptor tyrosine kinase* (*tek*) was identified as a statistically significant marker. The ortholog of *tek* in *S. purpuratus* is expressed in coel-omocytes and is upregulated in the apical organ during an immune challenge in adults (*Stevens et al., 2010*), indicating an immune function in sea urchins. We confirmed that these genes are expressed in mesenchymal populations using WMISH (*Figure 6C-E*). Double FISH (*Figure 6H.i-ii*) demonstrates that *tek* and the known-mesenchymal marker *ets1* are both expressed in mesenchymal populations (*Figure 6H*). We also identified *GATA binding protein 3* (*gata3*) as a marker of this cluster. Previous work in sea urchins has shown *gata3* plays a key role in blastocoelar immunocyte migration and maturation (*Pancer et al., 1999*). The WMISH presented in *Figure 6F* confirms that this gene is similarly expressed in patches of the mesodermal archenteron. This is consistent with marking pre-mesenchymal cells which have yet to ingress.

*elavl1-b* (*Jarvela et al., 2016*; *Hinman and Davidson, 2007*; *McCauley et al., 2010*) and *srcr* are also expressed in this cluster. A homolog of this *srcr* in the asteroid *Asteria pectinifera* has been directly linked to larval immune response (*Furukawa et al., 2012*).

Genes linked to immune function in other animals, including *dual oxidase maturation factor 1* (*duox1*) (*Rada and Leto, 2008*), *ras-related protein Rap-2c* (*rap2c*) (*Gillespie et al., 2022*), and *Vitamin D3 Receptor-like* (*vdrl*) (*Newmark et al., 2017*), were also detected as markers of this cluster.

We also identified *vascular endothelial growth factor receptor 1-like* (*vegfr1l*) as a marker of Pmin CL 21. WMISH of this gene (*Figure 6G*) shows it localized to the mesodermal cells at the top of the archenteron and in a mesenchymal population that accumulates around the base of the archenteron, similar to the localization of skeletogenic mesenchyme in *S. purpuratus* (*Duloquin et al., 2007*).

## Coelomic clusters

Two clusters, Pmin CL 14 and 16, were annotated as coelomic mesoderm (*Figure 7A*). *P. miniata* larvae have large coelomic pouches (*Figure 1*). These are formed from the mesoderm at the blastula stage, which buds off from the top of the archenteron during late gastrulation, and finally diverges into morphologically distinct left and right pouches. The left coelomic pouch will form the adult water-vascular system after metamorphosis (*Child, 1941*). The fate of the right coelom following metamor-phosis in *P. miniata* is unknown, but has been linked to the development of the adult pericardium in

other echinoderms (*Ezhova and Malakhov, 2021*; *Ezhova et al., 2016*). Further work is needed to confidently deduce the fate of the right coelomic cells in adults.

To examine the similarity between the marker gene sets of these two coelomic clusters, we calculated the representation factor (RF), a measure of the significance of the overlap between two sets of objects, compared to what would be expected by chance. Values greater than 1 indicate greater-than-expected overlap between two sets, while values less than 1 indicate less-than-expected overlap. These two clusters share 13 marker genes in common, with a representation factor value of 50.2 and p-val <2.921e-19, representing a high degree of similarity. Many of those shared genes are related to the extracellular matrix or cell/cell adhesion, such as *collagen alpha-2(IV) chain-like (col4a2l)*, *fibulin-1*, *collagen type IV alpha 1 chain (col4a1)*, *mucin-16-like (muc16)*, *teneurin-3-like (tenm3l)*, and *signal peptide, CUB and EGF-like domain-containing protein 1 (scube1)*. Curiously, there are few regulatory genes shared by these populations at this time point, with only *transcription factor AP4 (tfap4)* and *forkhead box p4 (foxp4)* in common. These overlaps suggest that both coelomic pouches share similarities in tissue structure yet have distinct regulatory profiles.

Pmin CL 16 is distinguished from Pmin CL 14 by the differential expression levels of *paired box 6 (pax6)*, *GATA binding protein 6 (gata6)* and *msh homeobox-like (msxl)* (*Figure 7B*). *gata6* (*Figure 7D*) is expressed more broadly in the left coelomic pouch but is absent from the tip of the right coelomic pouch. WMISH of these genes in later gastrula stages shows that *pax6* (*Figure 7E*) and *msxl* (*Figure 7F*) are at the tip of the left coelomic pouch and absent from the right. Therefore we annotated Pmin CL 16 as the Left Coelom.

*gata3*, *vegfr1l*, and *ets1*, which were markers of the Immune-like Mesenchyme, Pmin CL 21 (*Figure 5B*), were also found to be highly expressed in Pmin CL 14, relative to Pmin CL 16. WMISH revealed strong localization of *vegfr1l* and *gata3* to the right coelomic pouch (*Figure 7G-H*). *AT-rich interactive domain-containing protein 3 A (arid3a*, formerly referred to as *protein dead ringer homolog*) was also identified as a marker of this right coelomic region. *arid3a* has been shown to be involved in the epithelial to mesenchymal transition (EMT) and primary mesenchymal formation

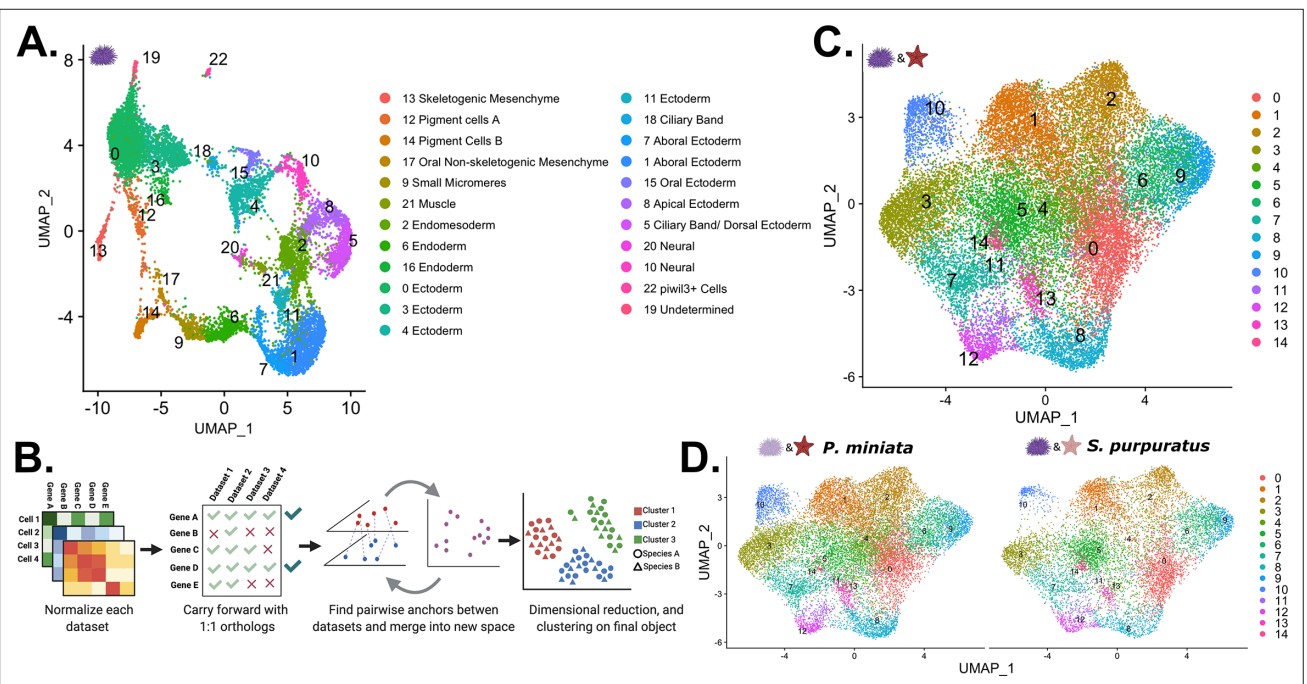

**Figure 8.** Creation of a developmental atlas in *S. purpuratus* and integration into a multi-species atlas. (**A**) UMAP projection of the 23 clusters identified and annotated in our *S. purpuratus* dataset used in later analysis. (**B**) Our pipeline for creating a multi-species atlas that uses only 1:1 orthologs to integrate the datasets into one common space using CCA. (**C**) The UMAP reduction of our multi-species atlas with 15 identified clusters. (**D**) Views of only the *P. miniata* (left) and *S. purpuratus* (right) data points in the multi-species atlas.

The online version of this article includes the following figure supplement(s) for figure 8:

**Figure supplement 1.** Differential expression of marker genes used to annotate *S. purpuratus* atlas.

in other echinoderms (*Amore et al., 2003*). The expression of these genes associated with mesenchymal transitions suggests that the right coelomic cluster is the point of origin of mesenchymal cells. Additionally, we found that transcription factor *AP-2 alpha* (*tfap2a*), also acts as a marker for, and transcripts localize, to the right coelomic pouch (*Figure 7I*).

For the time points examined in this paper, there is no noticeable morphological distinction between left and right of the *P. miniata* embryo, however there are clear transcriptomic divergences before left/right symmetry is morphologically observable.

## *S. purpuratus* single-cell atlas from equivalent stages

In order to compare *P. miniata* to *S. purpuratus*, cell cluster identities, an integrated atlas was necessary. An extensive comparison of whole embryo RNA-seq time series data between *S. purpuratus* and *P. miniata* has shown that *P. miniata* develops at a faster rate and that *S. purpuratus*, hatched blastula (12–15 hpf), mesenchymal blastula (18–20 hpf), and early gastrula (24 hpf) stages, are equivalent in developmental timing to the *P. miniata* time points used to generate the atlas here (*Gildor et al., 2019*). We, therefore, constructed an *S. purpuratus* atlas using samples taken at these stages from a previously published dataset (*Foster et al., 2020*). Our pilot snRNA-seq data was not used for the multi-atlas construction, as it was only a small exploratory data set. After quality control, 10,611 cells were selected for analysis and clustering. UMAP dimensional reduction and Louvain community assignment resulted in 23 distinct clusters, assigned the prefix Spur CL (*Figure 8A*). Clusters were annotated following the same marker gene method used for *P. miniata* (see Methods).

Below we discuss in greater depth the identification and annotation of *S. purpuratus'* Pigment Cell and Skeletogenic Mesenchyme clusters. These cell types are novel to *S. purpuratus* and thus of great interest in our comparison with *P. miniata*. We also document the identification and annotation of the Oral Non-skeletogenic Mesenchyme and stem/germ cell like piwil3+ lineage, as they are relevant in later discussion. *Figure 8—figure supplement 1* is a dot plot of genes used to annotate the other clusters in our *S. purpuratus* atlas, with those also used by *Foster et al., 2020* highlighted with an asterisk.

Pigment Cells (Spur CL 12 and Spur CL 14) and Skeletogenic Mesenchyme (Spur CL 13) were identified as distinct annotated clusters on the *S. purpuratus* UMAP projection (*Figure 8A*).

The Skeletogenic Mesenchyme cluster (Spur CL 13) was marked by the expression of regulatory genes such as *ets1, aristaless-like homeobox (alx1), ALX homeobox 1-like (alx1l,* previously called *alx4), T-box brain transcription factor 1 (tbr), ets variant transcription factor 6 (etv6,* previously called *tel), transcriptional regulator ERG (erg), hematopoietically-expressed homeobox protein HHEX homolog (hhex), TGFB induced factor homeobox 2-like (tgif2), arid3a,* and *vascular endothelial growth factor receptor 1 (flt1,* also known as *vegfr1)*. As expected, this cluster was also marked by genes involved in biomineralization, including *27 kDa primary mesenchyme-specific spicule protein (pm27a), spicule matrix protein SM50 (sm50),* and *mesenchyme-specific cell surface glycoprotein (msp130)*.

Two clusters were identified as belonging to the pigment cell lineage (Spur CL 12 Pigment Cells A and Spur CL 14 Pigment Cells B). These clusters shared 107 markers including known markers of pigment cells in sea urchins, including *gcml, polyketide synthase 1 (pks1), dimethylaniline monooxygenase [N-oxide-forming] 2 (fmo3),* and *sulfotransferase (sult)* (*Calestani et al., 2003*).(*Materna et al., 2013*) We hypothesize Spur CL 14 Pigment cells B represents a population of pigment cells in an earlier state of differentiation than Spur CL 12 Pigment Cells A, because upstream transcription factors such as *etv6, gata6, and etv1* mark only Spur CL 14 Pigment Cells B.

Another cluster of particular interest is Spur CL 17 Oral Non-skeletogenic Mesenchyme, the lineage fated to become immune coelomocyte cells. This cluster is marked by genes known to be involved in the specification of this lineage like *erg, gata3, gata6, hhex, prox1, ets1,* as well as genes that have been linked to immune response in echinoderms, like *tek,* and *srcr*. Other genes implicated in immune function in other systems such as two *allograft inflammatory factor 1-like* genes (LOC763226, LOC115929190), *C-type lectin domain-containing protein 141-like,* and *duox1* are also found in this cluster (*Brown et al., 2018*; *Rada and Leto, 2008*; *Vizioli et al., 2020*).

Spur CL 22 is annotated as piwil3+ Cells, because they are marked by the expression of *piwi-like RNA-mediated gene silencing 3 (piwil3)*. It is also marked by the gene *nanos C2HC-type zinc finger 2-like (nanos2l),* which has been shown to play a role in Primordial Germ Cell (PGC) specification (*Juliano et al., 2010*; *Yajima et al., 2014*). However, it must be noted that while this cluster could

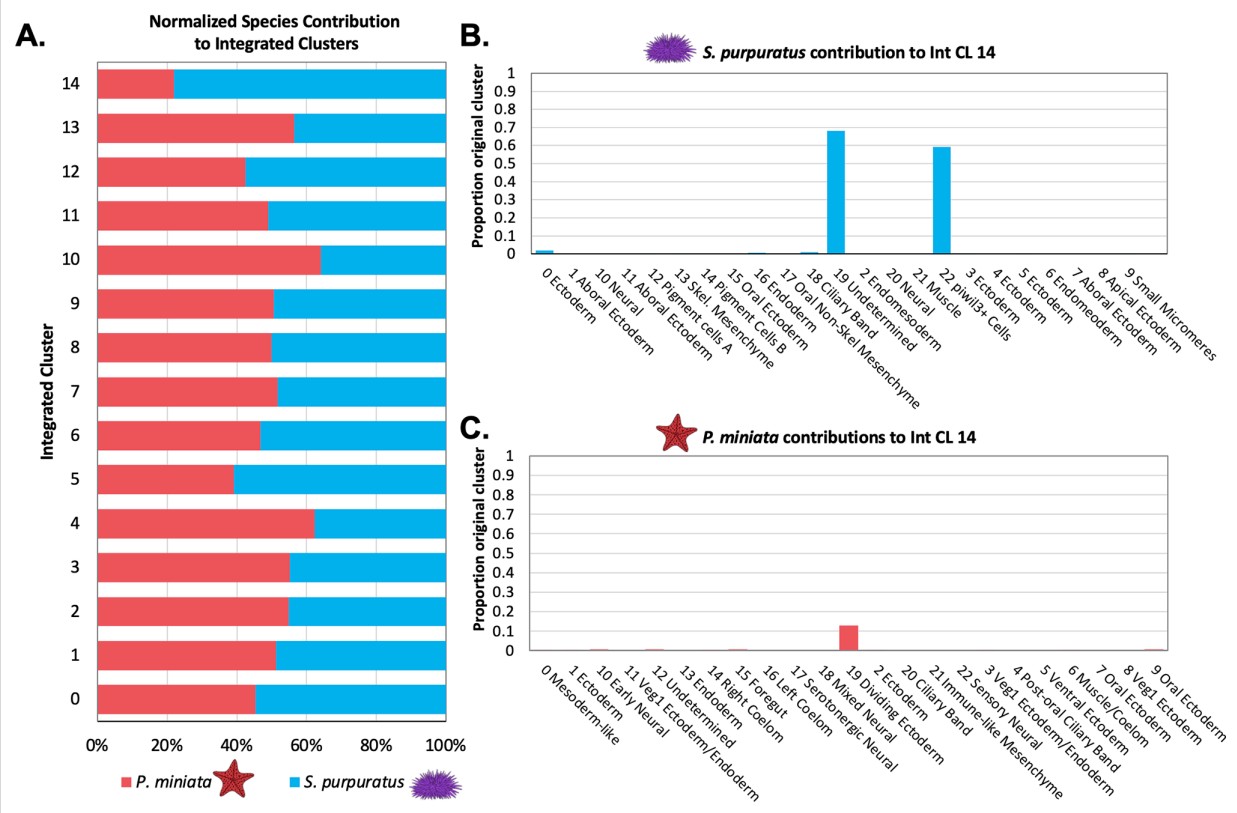

**Figure 9.** Quantification of species' contribution to integrated clusters and the identification of a cluster enriched for *S. purpuratus* cells. (**A**) Bar graph shows the percentage of nuclei/cells of each species in each integrated cluster, normalized to account for differences in sample sizes. Species contribution is balanced in all but Int CL 14. (**B**) A bar graph showing the percentage of each Spur CL # that contribute to Int CL 14. (**C**) A bar graph showing the percentage of each Pmin CL # that contributes to Int CL 14.

potentially be PGCs, further studies are needed to determine if this population is simply a *piwil3* positive group of cells with stem cell like properties or a genuine germ cell lineage.

## Multispecies integration

In order to directly compare sea star and sea urchin cell types, we projected the datasets from the two species into a single shared space (*Figure 8B*). We took advantage of extensive orthology analysis that has identified 1:1 orthologs between *P. miniata* and *S. purpuratus*, using a DIOPT-like system (*Foley et al., 2021*) and subsetted the datasets to only include genes that are 1:1 orthologs and are expressed in both datasets (see methods). This filtered gene set had 5799 genes (28.3% of *P. miniata*'s total genes, 21.4% of *S. purpuratus*' total genes), removing species-specific genes and genes with 1:many orthologs from our analysis. Normalization, integration, clustering, and dimensional reduction were done according to the same pipeline used in creating our single-species atlases and are detailed in the methods section. Our analysis resulted in 15 integrated clusters (henceforth referred to with the prefix Int CL ID; *Figure 8C-D*).

### Integrated clusters show similar patterns of gene expression between species

We first questioned whether any of the integrated clusters were over-represented by cells/nuclei from one species. After normalizing for the number of cells/nuclei per species sample, we calculated the percentage of species-specific cluster cells/nuclei in each integrated cluster (*Figure 9A*). For Int CL 0 through Int CL 14, each species contributes to at least a third of the total integrated cluster contents, except for Int CL 14, which is 92.9% *S. purpuratus* cells (A table listing all cluster contributors is available in *Supplementary file 3*). Therefore, we concluded that other than possibly Int CL 14, integrated

clustering was not driven solely by species differences, batch effects, or any technical variations in the sampling method.

To explore the compositions of these integrated groupings, we examined the contribution of the annotated clusters from our species-specific atlas. For each integrated cluster, we calculated the percentage of each original cell type (Spur CL ID, Pmin CL ID) present in the integrated cluster (Int CL ID). A full table of these values is available in *Supplementary file 3*.

## piwi3l+ Cells are an *S. purpuratus*-specific Integrated Cluster

Int CL 14 was the most species-specific integrated cluster. It had 210 cells total, making up 0.87% of the total dataset. This cluster was skewed towards *S. purpuratus*, whose cells made up 77.95% of the cluster as opposed to *P. miniata*, whose nuclei made up 22.0%. The *S. purpuratus* components of Int CL 14 primarily originate from Spur CL 19 Undetermined (68.2% of all cells from this original annotation) and Spur CL 22 piwil3+ Cells (59.3% of all cells of this original annotation). From *P. miniata*, 11 out of 23 clusters contribute nuclei to Int CL 14, with all but one contributing 5 or fewer nuclei. Pmin CL 19 Dividing Ectoderm contributes 34 nuclei, making up 12.8% of all cells of this identity.

We hypothesize that the low membership of *P. miniata* nuclei, which originated from low-level contributions from a diverse array of originally annotated clusters, in this integrated cluster indicates that there are no populations in *P. miniata* that closely resemble the transcriptome profile of *S. purpuratus*'s piwil3+ Cells and the undefined cluster. In the *S. purpuratus* cluster, Spur CL 19 Undetermined is dominated by the marker gene expression of cell cycle factors. Therefore, we think the expression of these factors drives clustering with both cells from Spur CL 22 piwil3+ Cells and Pmin Dividing Ectoderm in the integrated dataset. Therefore the presence of *P. miniata* nuclei in this cluster is likely driven by the impact of cell cycle genes, statistical noise rather than transcriptomic similarity to the piwil3+ Cells, thus making Int CL 16 the only *S. purpuratus*-specific transcriptomic profile present at this stage of development.

We hypothesize that this could be caused by the differences in the timing of PGC formation. Previous work has shown that the genes typically used to identify them, such as *piwi, nanos*, and *pumillo*, only begin to show cell-type-specific expression in *P. miniata* by the late gastrula stage when the posterior endocoelom forms (*Fresques et al., 2014*), a time point that lies outside our sampling window. Therefore, this cluster can be considered novel to *S. purpuratus* within our sampled window.

## *S. purpuratus* Pigment Cells share transcriptomic features with *P. miniata*'s Immune-like Mesenchyme and neurons

Despite the novelty of their phenotype, and their distinct clustering pattern in the *S. purpuratus* cell atlas, the pigment cell clusters (Spur CL 14 and 12) do not form a novel transcriptomic cluster in the multispecies integrated clustering. These cells mainly segregate into two integrated clusters, Int CL 8 and Int CL 12 (*Figure 10A*).

When comparing the marker genes between the two pigment cell clusters and *P. miniata*'s Immune-like Mesenchyme, we see 12 1:1 orthologous marker genes shared between them (*Figure 10D*). To examine the statistical significance of the overlap in marker gene sets, we calculated the representation factor based on the intersection of species-specific atlas marker genes that were 1:1 orthologs with respect to the total number of 1:1 orthologs. *P. miniata* Immune-like Mesenchyme and *S. purpuratus* Pigment Cells A and B have a statistically significant overlap in markers, with a representation factor of 5.2 and corresponding p-value <2.343e-06.

Amongst these markers were some shown to be involved in immune function in other animals. This gene set includes *srcr, rap2c, c-Maf inducing protein (cmip)*, and *calumin* (*Gillespie et al., 2022*; *Liu et al., 2011*; *Smith et al., 2018*). We therefore predict that *S. purpuratus* Pigment Cells and *P. miniata* Immune-like Mesenchyme share conserved immune response elements. The common immune function of these clusters is further supported when considering 92.1% of Spur CL 17 Oral Non-Skeletogenic Mesenchyme also segregates into Int CL 12 and share 32 marker genes in common, including *gata3, tek, duox1, elk1, ets1*, and *mitogen-activated protein kinase 1 (mapk1)*.

Int CL 8 is made up of 40.8% of all cells originating from Spur CL 12 Pigment A and 31.0% of all cells originating from Spur CL 14 Pigment B (*Figure 10A*). The *P. miniata* cells contributing to this cluster primarily originated from Pmin CL 10 Early Neural (48.3%), Pmin CL 18 Mixed Neural (95.9%), and Pmin CL 22 Sensory Neural (87.2%) (*Figure 10B*).

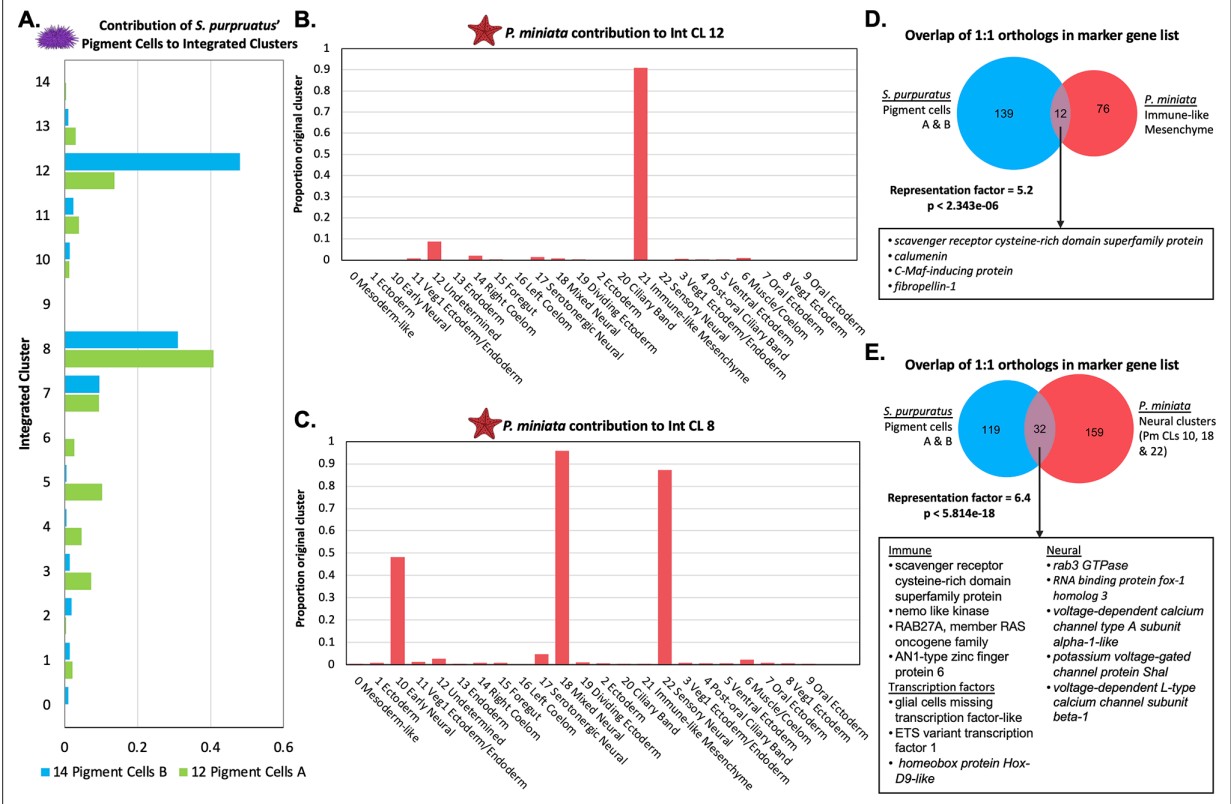

**Figure 10.** Pigment cells share transcriptomic similarities with *P. miniata* immune-like cells and neurons. (**A**) The proportion of total pigment cells from the *S. purpuratus* atlas (Spur CL 14 Pigment Cells B and Spur CL 12 Pigment Cells A) that contribute to each Int CL. The pigment cells chiefly contribute to Int CL 8 and Int CL 12. (**B**) A bar graph showing the percentage of each Pmin CL # that contribute to Int CL 12. *P. miniata* cells in Int CL 12 primarily originate from Pmin CL 21 Immune-like Mesenchyme. (**C**) A bar graph showing the percentage of each Pmin CLs that contribute to Int CL 8. The *P. miniata* cells in Int CL 8 primarily originate from three neural clusters: Pmin CL 10 Early Neural, Pmin CL 18 Mixed Neural, and Pmin CL 22 Sensory Neural. (**D**) A Venn diagram highlights marker genes that are shared between the Spur CL Pigment Cell clusters and Pmin CL Immune-like Mesenchyme. The overlap is higher than expected by chance, with a representation factor of 5.2 and p-value <2.343e-06. Highlighted in the box are shared genes with known immune function. (**E**) A Venn diagram shows the overlap of marker genes shared between Spur CL Pigment Cells and the pooled marker genes of Pmin CL 10 Early Neural, Pmin CL 18 Mixed Neural, and Pmin CL 22 Sensory Neural. The representation factor of this overlap is 6.4 and p-value <4.814e-18. Highlighted in the box are shared marker genes, divided into three categories: genes known to be involved in neural function, genes known to be involved in immune function, and transcription factors.

Comparisons of the marker gene sets of the Pigment Cells and merged Neural cells (excluding serotonergic cells), identified 32 gene orthologs in common, with a representation factor of 6.4 and corresponding p-value <5.814e-18 (*Figure 10E*). *S. purpuratus* Pigment Cells A and B and *P. miniata* Non-Serotonergic Neurons had several transcription factors shared as marker genes: *hoxd9a*, *etv1*, and *gcml*. In *S. purpuratus*, *gcml* is a known marker of pigment cells (*Calestani et al., 2003*; *Ransick and Davidson, 2012*). Several genes related to neuronal function were also identified as shared markers including *rab3*, *voltage-dependent calcium channel type A subunit alpha-1*, *potassium voltage-gated channel protein shal*, and *voltage-dependent L-type calcium channel subunit beta-1*. *P. miniata* neurons and *S. purpuratus* pigment cells also share marker genes potentially involved in immune function, including *srcr*, *ras-related protein Rab27-A*, *AN1-type zinc finger protein* and *nemo-like kinase (nlk)* (*Lv et al., 2016*).

## *S. purpuratus*' Skeletogenic Mesenchyme cluster with *P. miniata*'s Foregut and Right Coelom

Spur CL 14, annotated as Skeletogenic Mesenchyme, does not form a distinct transcriptomic cluster in the multi-species integration map (*Figure 11A*). However, 53.1% of all Skeletogenic Mesenchyme segregate into Int CL 7, while the rest are scattered across different integrated clusters.

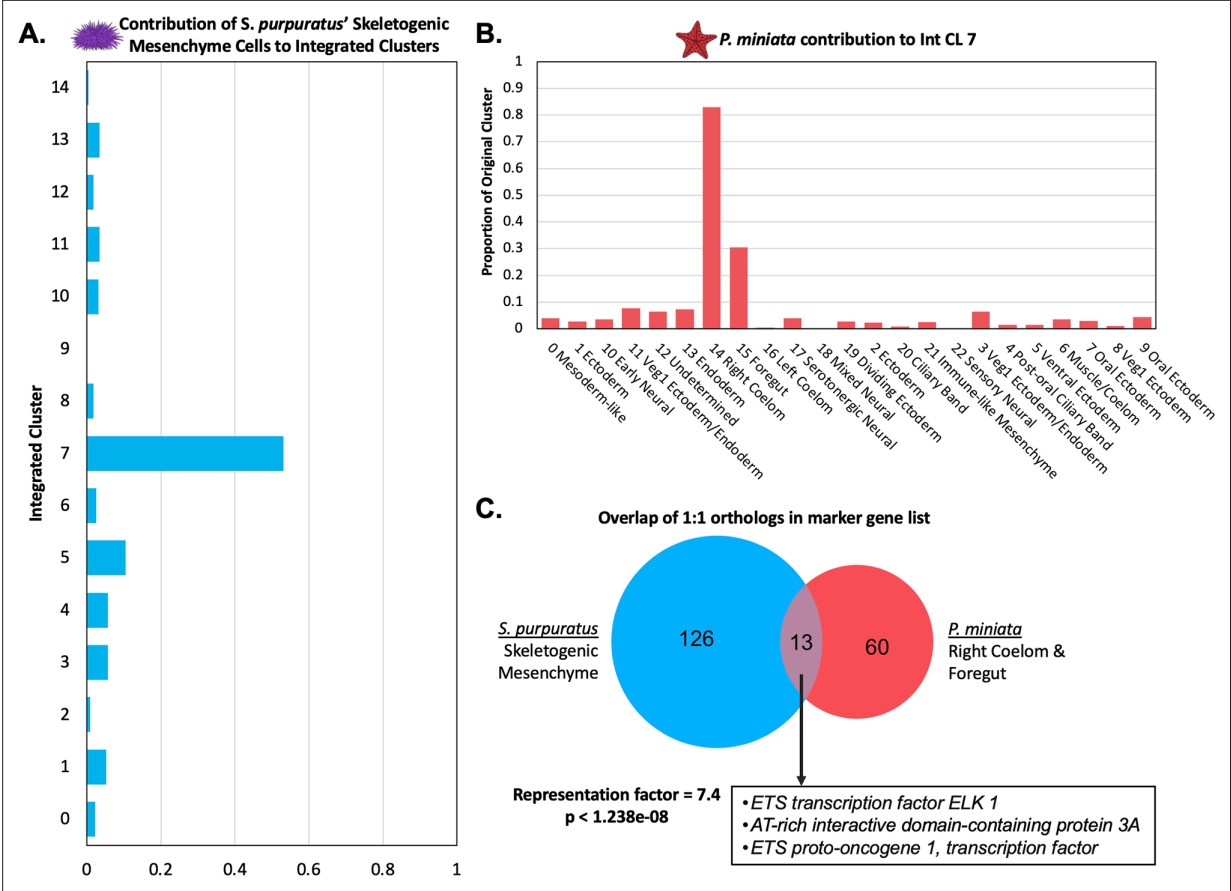

**Figure 11.** Skeletogenic Mesenchyme share transcriptomic features with *P. miniata*'s Right Coelom and Foregut. (**A**) The proportion of all cells annotated as skeletogenic mesenchyme from the *S. purpuratus* atlas that contribute to each Int CL. 53.1% of all skeletogenic mesenchyme cells segregate into Int CL 7. (**B**) A bar graph showing the percentage of each Pmin CL # that contributed to Int CL 7. The top contributors are nuclei from Pmin CL 14 Right Coelom and Pmin CL 15 Foregut. (**C**) Venn diagram highlights the overlap of marker genes that are 1:1 orthologs that are shared between *S. purpuratus*' Skeletogenic Mesenchyme and *P. miniata*'s Right Coelom and/or Foregut. The representation factor is 7.4 with p-value <1.238e-08. Highlighted in the box are transcription factors shared by both groupings.

When we examine *P. miniata* cell types contributing to Int CL 7, the top contributors are Pmin CL 14 Right Coelom (83.0%) and Pmin CL 15 Foregut (30.5%) (*Figure 11B*). A comparison of *P. miniata's* Foregut markers to *S. purpuratus'* Skeletogenic Mesenchyme markers yielded a representation factor of 5.0 with a p-value <0.008 while the overlap of markers between *P. miniata's* Right Coelom and the Skeletogenic Mesenchyme resulted in a representation factor of 10.2 and p-value <1.819e-10 (*Figure 11C-D*). The three genes shared across all three original clusters are *fibrillin-3, sprouty-related EVH1 domain-containing protein 3 (spred2)*, and *3 alpha procollagen precursor (col4a1)*. Probable *JmjC domain-containing histone demethylation protein 2 C (hnhd1c)* acts as a marker in only *S. purpuratus'* Skeletogenic Mesenchyme and *P. miniata* Foregut.

In total, 13 marker genes are shared between the Skeletogenic Mesenchyme and *P. miniata's* Right Coelom. Of particular interest are the shared markers, *arid3a*, transcription factors *elk1*, and *ets1*, genes previously characterized in the skeletogenic mesenchyme GRN of *S. purpuratus* (*Davidson et al., 2002b*; *Rho and McClay, 2011*). This suggests there is a region of *P. miniata* that directly corresponds to skeletogenic mesenchyme, despite morphologically lacking them.

## *P. miniata* Left Coelom has no clear equivalent in early *S. purpuratus* gastrulation

Unlike the Right Coelom, there is little overlap in marker gene content between the Left Coelom and Skeletogenic Mesenchyme. These two groups share 4 marker genes in common *solute carrier family 4*

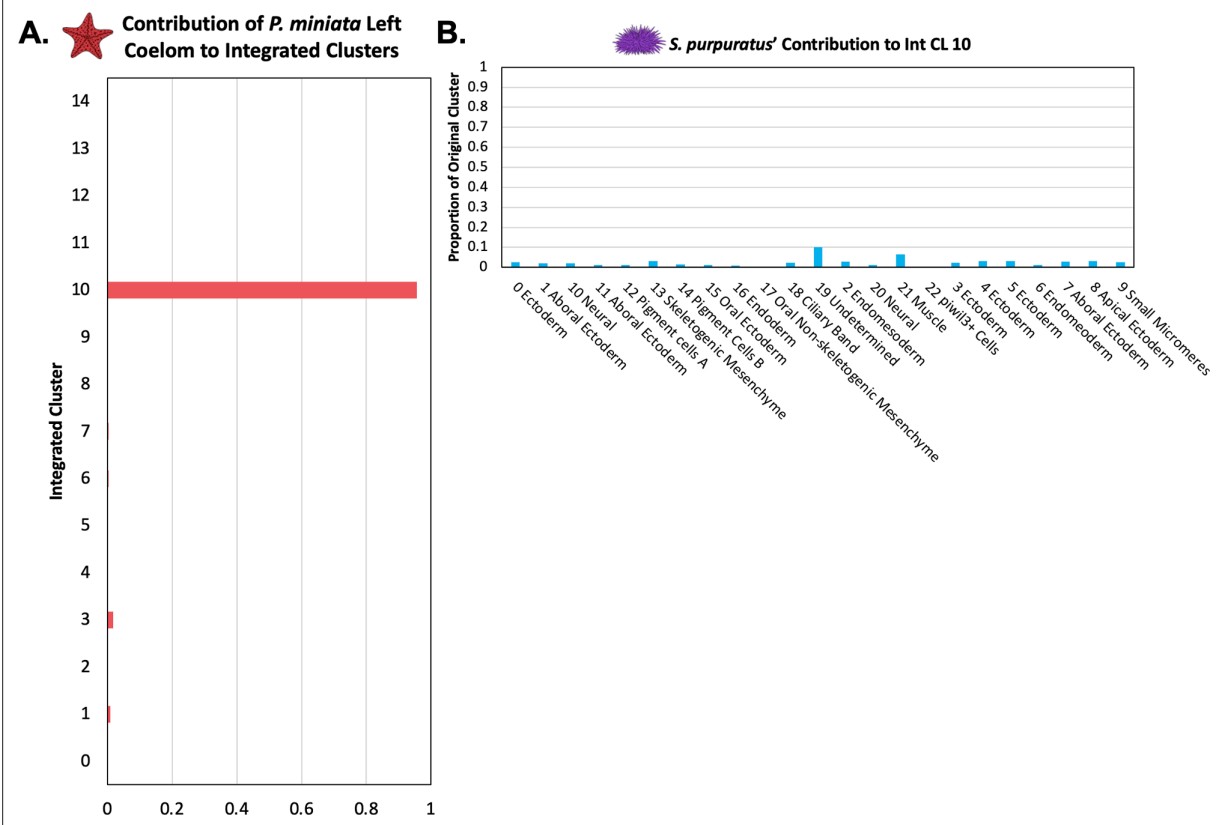

**Figure 12.** The *P. miniata* Left Coelom does not have a corresponding population in *S. purpuratus*. (**A**) The proportion of total nuclei originating from Pmin CL 16 Left Coelom across all integrated clusters. Most of the nuclei are located in Int CL 10. (**B**) A bar graph showing the percentage of each Spur CL # that contributed to Int CL 10. The cells in Int CL 10 originate from many different Spur CL # and no single cluster contributes more than 10.2% of a cluster's original total cells.

member 11 (*slc4a11*), 3 alpha procollagen precursor (*col4a1*), rap associating with DIL domain (*radil*), and *DENN domain-containing protein 5B* (*dennd5b*) with a representation factor of 6.3 and p-value <0.0033.

We, therefore, were especially interested in comparing the cluster identity of the Left Coelom. 95.7% of Pmin CL 16 Left Coelom nuclei contribute to Int CL 10 (*Figure 12A*). When we examine the contributions of *S. purpuratus* cells to that cluster, we see that there is no main contributor, rather small contributions across a wide range of *S. purpuratus* cell types, originating from 21 of the 23 different Spur CL clusters, with contributions ranging from 0.08% to 2.9% of a given Spur CL.

There are several potential explanations for the isolation of the left coelom cells in our integrated clustering. Firstly, this could correspond to a novel cell lineage that has not been documented in sea stars. Alternatively, the transcriptomes of the homologous coelomic pouches between the species could be sufficiently divergent so that they no longer drive clustering in the integration approaches we used, possibly reflecting to the morphological uniqueness of sea star coeloms with respect to other echinoderm species (*Child, 1941*; *Morris et al., 2009*). This could also, like with the piwil3+ Cells, result from a difference in developmental timing in which the corresponding population has not yet arisen in *S. purpuratus* at the sampled stages. Further interrogation of *P. miniata's* left coelomic pouch development is needed to better understand these phenomena.

## Discussion

At the heart of evolutionary developmental biology rest two questions: what is a cell type? And what is novelty? Being able to differentiate and reproducibly define cell categories is essential to understanding the evolutionary relationship between cell types and the cross-species comparisons needed to establish novelty. The identification and definition of cell types can change with new technologies

as new data is brought to light. In this paper, a single-nucleus RNA-sequencing approach was used to classify cell types present during early embryonic development in the bat sea star, *P. miniata*. An integrated atlas was constructed using 1:1 orthologs of *P. miniata* and *S. purpuratus* to investigate and infer the changes during cell type specification that underlie the developmental origins of these novel cell types.

## An early embryo developmental atlas for *P. miniata*

We produced a developmental atlas of an echinoderm belonging to the Asteroidea family. The expected cell types were observed and known markers of major tissue types were identified (summarized in *Figures 3 and 4*). We were also able to make several key observations of cell types present in sea star embryos for the first time. We observed 4 neural clusters in our analysis. These include serotonergic neurons, sensory neural, mixed neural, and an early, general neural type. The identification of these distinct neural clusters is consistent with the finding that the sea urchin early gastrula embryo has three neural cell types (*Slota and McClay, 2018*). In sea urchins, these three neural subtypes are distinguished by the expression of unique transcription factors. The work here shows that neural subtypes can also be distinguished by the expression of different synaptotagmin genes in *P. miniata*, suggesting distinct synaptic vesicle trafficking in these neural types. The identification of endocrine-associated markers in two of these neural clusters (Pmin CL 18 Mixed Neural and Pmin CL 22 Sensory Neural) supports an emerging hypothesis of the origins of neural subtypes from secretory cells (*Moroz, 2021*), which is thus still evident in the mixed endocrine and neural markers of these clusters. This is also consistent with recent work in *S. purpuratus* larvae, which identified endocrine-associated genes in post-oral/lateral neurons and argues that the endocrine system evolved from a subset of neurons (*Paganos et al., 2021*; *Perillo et al., 2018*). The similarities found in sea stars and sea urchin embryos demonstrate that the striking degree of neural diversity is a likely feature of early eleutherozoan embryos.

Diverse mesodermal cell types were also identified by snRNA-seq in *P. miniata*, including an immune-like population (*Figures 4 and 6*). In the *P. miniata* Immune-like Mesenchyme cluster were orthologs of the sea urchin blastocoelar and pigment cells' immune response gene *tek*, a gene expressed in sea urchin coelomocytes and linked to the proliferation of immune cells following an immune challenge (*Stevens et al., 2010*). *P. miniata* Immune-like Mesenchyme and sea urchin blastocoelar cells also share the marker genes *duox1* and *mapk1*, which are well documented to play a role in stress and immune response (*Pinsino et al., 2015*; *Smith et al., 2018*; *Zhan et al., 2018*). These genes are not detected as markers in the sea urchin Pigment Cell cluster, reinforcing the idea that the pigment and blastocoelar cells play different roles in the immune response (*Ch Ho et al., 2016*).

The two populations of coelomic mesoderm in *P. miniata* are of special interest. The Right Coelom shares many markers with the Immune-like Mesenchyme cluster (including transcription factors such as *elk1* and *ets1*, *Figures 4 and 6*), suggesting that the Immune-like Mesenchymal cells originate from the right coelom and may continue to do so throughout development. However, the left coelom does not have mesenchyme markers suggesting that cells in the left coelom do not undergo EMT transition at this developmental stage. This asymmetry is maintained throughout later developmental stages outside of our sampling window (e.g. shown by whole-mount in situ hybridization in *Figure 7D-I*).

## A comparative single nuclei/cell transcriptomic approach for studying the evolution of cell types

Identifying homologous cell types across species is a central quest in biology as it is the prerequisite knowledge for understanding the evolutionary basis of cell and morphological diversification. Here, we have presented an orthology-based methodology for combining developmental single cell/nuclei data across species into one composite atlas. Current approaches focus on either integrating data from multiple species into a common atlas or performing pairwise comparisons between separately clustered datasets (*Shafer, 2019*; *Tanay and Sebé-Pedrós, 2021*). An advantage of examining clustered multi-species transcriptome data is that it accommodates split clustering of an organism's original cluster identity, where a transcriptomic profile defined in a species-specific developmental atlas may split between more than one cluster into the integrated data. This allows us to pull out elements of the GRN that overlap, which we can combine with biological knowledge to form hypotheses for the sister cell relationships and the identification of shared modules deployed

in different contexts. We have shown that using only 1:1 orthologs to create an integrated atlas is sufficient to drive clustering that results in mixed-species clusters that demonstrate meaningful transcriptomic similarity. However, it should be noted that restricting our analysis to only 1:1 orthologs removes data from an analysis that can potentially mask the importance of non-orthologous genes and gene duplications in driving species-specific cell-type differences. As this field advances, integration accounting for paralogous transcripts could help improve the meaningfulness and depth of transcriptome comparisons.

To explore the concept of cell novelty, we first examined whether any integrated clusters showed a species bias. If a cell type was completely unique to one species, we would expect to see a cluster on our integrated UMAP that is dominated by contributions from one species. Int CL 16 is our most species-skewed cluster, with 7.1% from our *P. miniata* dataset and 92.9% of its members originating from our *S. purpuratus* dataset, most of which were originally annotated as piwil3+ Cells (*Figure 10*). The sea urchin piwil3+ Cells could therefore be viewed as novel under the GRN definition of cell types as they do not share any significant transcriptional similarity with cell clusters in sea stars. However, an alternative interpretation is that these cells have yet to be specified in the stages of our sampling in sea stars. There is evidence that these cells are specified later in sea stars (*Fresques et al., 2014*). Therefore, we propose that piwil3+ Cells can likely be considered temporally novel to *S. purpuratus* in our dataset, with its distinct signal arising from differences in developmental timing and not separate evolutionary cell lineages. This distinction can be resolved by expanding the datasets to later time points and establishing whether the piwil3+ Cells would form mixed-species clusters.

Our analysis also identified another potentially species-specific cell population, the Left Coelom of *P. miniata*, which primarily contributed to Int CL 14 (*Figure 10*). Though it was not species-exclusive, the *S. purpuratus* atlas of the cells that contributed to the integrated cluster were of numerous identities with no clear contribution from any cluster in *S. purpuratus*. In both species, the left coelomic pouch gives rise to the hydrocoel, which forms the adult water vascular system in adults and is a shared feature of all echinoderms (*Balser et al., 1993*; *MacBride, 1918*). Therefore from a functional and developmental perspective, these cells are not considered novel but using our definition of transcriptome similarity, they may represent a novel population in sea stars. Understanding the GRN operating in this coelom will provide further insights into the role of this unique regulatory transcriptome. This in turn may change the understanding of the origins of the water vascular systems which is an echinoderm synapomorphy.

In comparison, the presumed, traditionally defined novel cell types, that is the skeletogenic mesenchyme and pigment cells in sea urchins form molecularly distinct populations in the single species atlas but lose that distinctness in our multi-species atlas where we find similarities between their transcriptomes and those of *P. miniata* clusters. In our integrated data set, the Skeletogenic Mesenchyme aggregate with nuclei from *P. miniata*'s Right Coelom, and these two populations share a significant number of marker genes (*Figure 12*), including several transcription factors (*ets1, arid3a, elk1*). We suggest that these genes have a conserved role in the core mesenchyme specification program present in their last common ancestor. Further work examining these cell types across species of echinoderms is needed to better define these novelties and allow for a more comprehensive characterization of their evolutionary timing.

Of particular interest is our detection of *vegfr1l* in *P. miniata*. Previous studies have shown that an ortholog of this gene, *vegfr1l* (also known as *flt1*) was exclusively expressed in the skeletogenic mesenchyme of brittle stars and sea urchins and was responsible for the deployment of the biomineralization gene set as was seen as a critical node of cooption (*Morino et al., 2012*). VEGFR signaling does not activate skeletogenesis pathways, at least at this developmental stage, as the sea star embryo does not make a skeleton. The *P. miniata* skeletogenic mesenchyme-like GRN now identified in the right coelom may instead have a conserved role in cell guidance, and/or establishing the clustered ring of primary mesenchymal cells at the bottom of the blastocoel, as it does in skeletogenic mesenchyme (*Morino et al., 2012*). In *Figure 6G*, we see *vegfr1l*-positive cells form a ring around the base of the archenteron, similar to skeletogenic PMC cells. This is an intriguing finding as it suggests that the patterning and migration cues that direct skeletogenic mesoderm cells in sea urchins are also in place in sea stars. This indicates that perhaps only the very final activation of the skeletogenesis program is present in sea urchins and that otherwise the cell lineage between these species is similar. This finding is also consistent with the findings from Cidaroid sea urchins that show that skeletogenic cells arise

from a population of mesenchyme that migrates from the top of the archenteron (*Erkenbrack and Davidson, 2015*).

*Vegfr* has also been shown to control conserved tubulogenesis programs documented in other echinoderms and vertebrates (*Ben-Tabou de-Leon, 2022*; *Morgulis et al., 2019*). The coelomic pouches of asteroids are very distinct from other species of echinoderms; they form much earlier in embryonic development and are longer and thinner, compared to other echinoderms (*Child, 1941*). Therefore, we hypothesize that some of those conserved tubulogenesis genes may also be involved in developing their distinct coelomic morphologies while in sea urchins this gene is instead used to activate biomineral pathways.

In our multi-species atlas, pigment cells that formed a cluster in the *S. purpuratus* atlas, contribute to integrated clusters with *P. miniata* immune-like mesenchyme or *gcml*-positive neurons. It is not surprising that pigment cells may share similarities with immune cell types in sea stars, given their presumed shared immune functions. When comparing the immune-like mesenchyme of the sea star with the pigment cells of *S. purpuratus* we see several shared combinations of transcription factors, previously demonstrated to play similar roles in cell fate specification between *S. purpuratus* and *P. miniata*. The sharing of these early-acting regulators suggests that there is a common mesodermal and immune-like mesenchyme regulatory program maintained between *S. purpuratus* and *P. miniata*. This supports previous work hypothesizing the close evolutionary relationship and possible functional ancestral interdependence between the immune and nervous systems (*Klimovich and Bosch, 2018*; *Kraus et al., 2021*).

The pigment cell clustering with *P. miniata* neurons is consistent with previous work that has shown that in larval scRNA-seq, pigment cells cluster with ectodermal cell populations (*Paganos et al., 2021*). When comparing the gene expression patterns of the pigment cells and *P. miniata* neurons, we see an overlap of several genes. Of particular interest is their shared expression of *gcml*. Previous literature showing *gcml* is expressed in neural development in other phyla (*Jones et al., 1995*; *Soustelle et al., 2007*) leads us to suggest that *gcml* plays a role in the development of neural-like excitable or secretory cells in echinoderms, and in the common ancestor of the eleutherozoa. In this scenario, pigment cells may have arisen through the co-option of a neuronal *gcml* module in a subset of mesenchyme cells. The regulatory independence of this new *gcml*-expressing population allowed for genetic individualization and the evolution of different downstream functionalities including the activation of pigment formation genes. At the same time, we see the maintained expression of other neuronal effector genes (*thrb*, voltage-gated channels, and *rab3*), possibly another consequence of *gcml* expression. This suggests that these genes may play dual roles in immune and neural cells and the emergence of modern pigment cells is the result of a fusion of regulatory programs from different cell lineages. Additionally, work comparing fully differentiated neurons, immune cells and pigment cells would help elucidate any further similarities in cell function and gene expression, possibly providing insights into how novel cell types arise and develop unique phenotypes via the co-option of regulatory pathways.

## Conclusion

We have used single nuclei RNA-seq to study early embryo development in the sea star *P. miniata* and an orthology-centered approach to integrate it with the atlas of sea urchin, *S. purpuratus*. Single-cell transcriptomics provides a very detailed picture of gene expression and cell identity. This allows us to deeply explore concepts of cell-type identity and evolutionary novelty through the lens of gene expression and inferred GRNs operating in the cells.

The examples uncovered in this study serve to expand our understanding of evolution of cell types among these echinoderms, and more generally highlight ways that we can think about novelty and the limitations of experimental approaches to this understanding.

Our comparisons discovered examples of transcriptomically distinct cell clusters, that is piwil3+ Cells in sea urchins and Left Coelom in sea stars, with no clear orthologous cluster in the other species. These findings were unexpected and have identified previously unappreciated differences in cell types. These findings may be taken as examples of a novel cell when defined on the basis of gene expression similarities achieved by this method. Novelty in both of these cases may be relative changes in timing or heterochrony of the cell state specification. Unless snRNA-seq covers a tight time course across the entirety of development and adult tissues, and possible environmental contexts we can never truly determine novelty. However, heterochronic activation of a GRN in a new context can

result in dramatic functional differences to the organism. The very early specification of a *piwil3+* cell population in sea urchin likely points to a different developmental program and possible functional role of these cell types in these species.

Conversely, the two cell types that we anticipated as novel based on cell morphology, function, and developmental lineage, that is skeletogenic mesenchyme and pigment cells in sea urchins had significant transcriptomic profiles in common with sea star cell clusters. This implies a shared ancestry of these cells, with an ancestral cell type possessing these regulatory profiles. In the case of skeletogenic mesenchyme and right coelom, this may include overlapping functions in EMT, cell migration, and tubulogenesis with a cooption, quite late in the developmental GRN of cassettes of biomineral genes. The most surprising find from our studies was the overlap between pigment cell cluster transcriptome and the immune and neural clusters in sea stars. This implies a more complex evolutionary scenario whereby programs from two cell types in sea stars have merged into one to produce a distinct cell morphology and function.

In conclusion, our study provides new interpretations of cell type novelty and new inferences of novel cell type evolution when considering comparisons of orthologous gene expression profiles over traditional approaches.

## Methods

### Embryo isolation and prep

Adult *P. miniata* and *S. purpuratus* animals were obtained by Marinus Scientific (Long Beach, California) and housed in 100-gallon tanks containing aerated artificial seawater (Instant Ocean, 32–33 ppt between 10 and 15°C). To induce spawning, adults were injected with 200 µM 1-methyl-adenine (*Kanatani, 1969*). Male and female gametes were harvested and mixed for 10 min to facilitate fertilization. Zygotes were filtered out using a 100 µm mesh filter and washed with artificial seawater to remove excess sperm. Cultures were then transferred to 4-gallon buckets and incubated at 13 °C, with constant gentle shaking to prevent settling.

### Whole-cell dissociation

The whole-cell dissociation protocol was adapted from *Barsi et al., 2014*. Prior to dissociation, whole embryos were washed and resuspended in 500 µL artificial seawater containing 2 mg/mL Pronase, incubated on ice for 3 min, and centrifuged at 700 $g$ for 5 min at 4 °C before being resuspended in ice-cold calcium-free artificial seawater (20 mg/mL: BSA). The embryos were then manually dissociated using a P1000 micropipette and filtered through a 30 µM mesh filter. The sample was then centrifuged at 300 g for 5 min at 4 °C, and resuspended in artificial seawater. A single sample was then processed for bulk RNA-seq (see below).

### Nuclear isolation

Prior to dissociation, embryos were washed twice with cold seawater by centrifuging for 5 min (22 g) at 4 °C, and resuspended in 10 mL cold HB Buffer (15 mM Tris pH7.4, 0.34 mM sucrose, 15 mM NaCl, 60 mM KCl, 0.2 mM EDTA, 0.2 mM EGTA, 2% BSA, 0.2 U/µL protease inhibitor (Sigma Aldrich cOmplete protease inhibitor)) and kept on ice. Washed embryos were then transferred to a 15 mL dounce, and homogenized 20 times with the loose pestle and 10 times with the tight pestle. The dounced sample was centrifuged for 5 min (3500 $g$) at 4 °C, and the pellet was resuspended in 1 mL HB buffer. Tip strainers (Scienceware Flowmi) were used to transfer the sample to a new tube and 9 mL ice-cold HB buffer was added to the sample and centrifuged for 5 min (3500 $g$) at 4 °C. The pellet was resuspended in 1 mL PBT (1 X PBS, 0.1% Triton X-100, 2% BSA, 0.2 U/µLl protease inhibitor [Sigma-Aldrich cOmplete protease inhibitor tablet], and 0.2 U/µL SUPERase-in RNAse inhibitor [Invitrogen]). Samples were centrifuged for 5 min (2.4 $g$) in a cold room and nuclei were resuspended in 300 µL PBT. Nuclei were stained with Trypan Blue and counted in triplicate with a Countess II Automatic Cell Counter (Thermo Fisher). The resulting nuclei were resuspended to 1000 nuclei/µL.

### Bulk RNA-seq sample preparation and sequencing

Bulk RNA sequencing was performed on 25 hpf *P. miniata* embryos that had been left whole, dissociated into single cells, or were nuclear isolates (*Figure 2*.A). Bulk RNA was isolated using the GenElute

Mammalian Total RNA Miniprep kit (Sigma). Illumina HiSeq1 Libraries were prepared for Illumina HiSeq 50 bp paired-end sequencing and sequenced on the NovaSeq 6000 (Duke Center for Genomics and Computational Biology, Durham, NC). Reads were aligned to the *P. miniata* genome version 2.0 (*Kudtarkar and Cameron, 2017*) using the standard TopHat pipeline (*Kim et al., 2013*), and gene counts were quantified using HTseq (*Putri et al., 2022*) and quantile normalized.

## Single nuclei library preparation and sequencing

Single nuclei libraries were created using the Chromium Next GEM Single Cell 3' Reagent Kit v3.1 RevD (10 X Genomics) and the Chromium X instrument (10 X Genomics) using manufacturer protocols. For *S. purpuratus*, the number of nuclei recovered was 2000 for the 6 hours post fertilization (hpf) sample, 5000 for the 15 hpf and 6000 for 23 hpf samples. For *P. miniata,* there were 3000 nuclei in the 16 hpf sample, 5000 in the 26 hpf sample, and 6000 in the 40 hpf sample. After constructing the libraries, samples were quantitated on a Tapestation using D5000 ScreenTape to confirm insert sizes of 400–500 bp. Samples were sequenced on a Novaseq 2X150 S4 lane (Duke Center for Genomics and Computational Biology, Durham, NC).

## Processing and quality control of snRNA-seq data

An outline of the code used to create both species atlases and the integrated atlases is available in the .

Reads were processed using Cell Ranger V4.0.0 (*Zheng et al., 2017*) and aligned to a pre-mRNA index based on the *P. miniata* genome V3.0 (GenBank assembly accession: GCA_015706575.1) and *S. purpuratus* genome V5.0 (GenBank assembly accession: GCA_000002235.4). The pre-mRNA indices were constructed using gene annotation files modified by an in-house script to remove intron annotations, allowing unspliced reads to align. All further analysis was conducted in R V4.0.5 (*R Core Team, 2021*).

Quality control for our nuclear datasets was carried out as described in *Luecken and Theis, 2019*, with thresholds for genes per cell and transcripts per cell set manually for each individual sample. The thresholds are summarized in *Supplementary file 2*. In *S. purpuratus* samples, we were able to introduce another quality control feature and remove nuclei that had more than 5% of their transcripts originating from genes located on the mitochondrial scaffold. The *P. miniata* mitochondrial scaffold has yet to be identified, and therefore we could not introduce this metric to its quality control. For the previously published datasets from *Foster et al., 2020*, we used the published thresholds (with the addition of an nCount_RNA threshold to less than 3,000 in the early gastrula dataset to remove potential doublets), with the addition of our 5% mitochondrial transcript limit.

## Creation of single-cell/nucleus atlas

Using Seurat V4.0.5 (*Hao et al., 2021*), each time point was individually normalized using the SCTransform algorithm (*Hafemeister and Satija, 2019*) while also regressing out the expression levels of rRNAs (as our library construction relied on polyA tagging). In our *S. purpuratus* samples, we also regressed out the percent mitochondrial transcripts. For the integration of individual samples, we used canonical correlation analysis (CCA) to identify up to 3000 genes best suited to anchoring the datasets together using the SCT normalized values (Note: only 2,313 anchors were identified in our multi-species atlas). Following integration, Principal Component Analysis was run on the composite object and significant Principle Components (PCs) were identified by plotting the standard deviation of the first 50 PCs and identifying the PCs that describe the most variability of the datasets.

Our *P. miniata* nuclei atlas used 20 PCs, our *S. purpuratus* nuclear atlas used 10 PCs, the *S. purpuratus* whole-cell atlas used 10 PCs, and our multi-species atlas used 10 PCs. Using our selected PCs, we constructed k-nearest neighbor graphs (with k=20), performed Louvian Community Assignment (resolution set as 2 for *P. miniata* data, 1.5 for *S. purpuratus* nuclei data, 1 for *S. purpuratus* whole-cell data, and 1 for multi-species data), and performed Uniform Manifold Approximation and Projection (UMAP) reduction in order to generate clusters. Resolution was determined for each species' dataset individually seeking to optimize the number of clusters, number of marker genes, distribution of marker genes between clusters, and maximum modularity with 10 random starts observed with different resolution.

## Calculating marker genes

To annotate cluster identity, we identified genes that differentially marked clusters using the Wilcox rank-sum test. A gene is considered a marker gene for a cluster if it is expressed in at least 5% of the cells in that cluster and has at least a 0.25 log fold-change difference in average expression between the cells in that cluster and all other cells. Adjusted p-values were calculated using Bonferroni correction and used to assess significance. We then used these marker genes to determine cluster cell-type labels by cross-referencing them with genes established in previous studies to have cell-type-specific expression.

## Ortholog identification and multi-species integration

1:1 orthologs were identified between *S. purpuratus* genome V 5.0 and *P. miniata* genome V3.0 and named according to orthologs identified in the human genome. For a full description of ortholog identification see *Foley et al., 2021*, which used a variety of tools (BLAST, InParanoid v4.1, ProteinOrtho v6, SwiftOrtho, FastOrtho, OMA and OrthoFinder) to identify orthologous relationships. To facilitate dataset integration, where applicable, the gene IDs in P. miniata datasets were converted to their orthologous S. purpuratus gene IDs. In total, 5799 of the 5935 1:1 orthologous genes identified between both species were expressed in at least 2 cells/nuclei in this dataset.

## Multi-species cluster composition and comparisons

To facilitate this comparison we constructed an *S. purpuratus* atlas at comparable time points to our *P. miniata* set, using the hatched blastula, mesenchymal blastula, and early gastrula stage data from a previously published dataset (*Foster et al., 2020*). CellRanger V 4.0.0 (*Zheng et al., 2017*) was used to align reads to a pre-mRNA index created using the *S. purpuratus* genome v5.0. After quality control (selection of transcript and gene count thresholds as with the *P. miniata* atlas, as well as the removal of cells with more than 5% of their transcripts originating from the mitochondrial scaffold), 10,611 cells were selected for analysis and clustering using Seurat V4.0.5 (*Hao et al., 2021*).

For each time point, reads were normalized using the SCTransform algorithm (*Hafemeister and Satija, 2019*), while regressing out mitochondrial (for *S. purpuratus* only) and rRNA transcripts. 2321 integration anchors were identified and used for dataset integration via canonical correlation analysis (CCA), using the Seurat package (*Stuart et al., 2019*). UMAP dimensional reduction and Louvain community assignment resulted in 20 distinct clusters.

To compare marker genes between different clusters, all calculated marker genes for a cluster of interest in their species-specific atlas were input into a Venn Diagram tool, housed by the Bioinformatics and Evolutionary Genomics group at the Ghent University (http://bioinformatics.psb.ugent.be/webtools/Venn/).

To calculate the significance of marker gene overlap between species, we calculated the representation factor of pairwise comparisons between calculated marker gene lists. It is calculated by dividing the number of overlapping genes by the expected number of overlapping genes for two independent sets subsampled from a larger finite set. Values over 1 indicate more overlap than expected by chance, while values under 1 indicate less overlap than expected. For marker comparisons between clusters in *P. miniata,* a total gene number of 27,818 was used. For cross-species analysis, this was calculated with respect to the total numbers of 1:1 orthologs. A calculator housed at http://nemates.org/MA/progs/representation.stats.html was used to conduct this analysis.

## Whole mount in situ hybridization (WMISH)

The WMISH protocol was adapted from *Hinman et al., 2003b*. Briefly, embryos were fixed in MOPS-4% PFA buffer and stored in 70% ethanol at –20 °C. In situ primers for genes of interest were designed using Primer3 (*Koressaar and Remm, 2007*). Sequences were amplified from cDNA and DIG-labeled RNA probes complementary to the target mRNA (Roche). The primer sequences for synthesizing the probes are available in *Supplementary file 4*. Three non-overlapping probes were created for *P. miniata's vegfr1l*.

Fixed embryos were brought to room temperature, placed into in situ buffer (0.1 M MOPS, 0.5 M NaCl, 0.1% Tween 20) washed twice, then transferred into hybridization buffer (0.1 M MOPS, 0.5 M NaCl, 1 mg/mL BSA, 70% formamide, 0.1% Tween 20) and pre-hybridized overnight at 58 °C. Following denaturation for 5 min at 65 °C, DIG-labeled probes were added to the embryos at 0.2 ng/µL. Samples

were then incubated at 58 °C for 4–7 days. Embryos were washed three times with a hybridization buffer over 4–6 hr, transferred to MAB (0.1 M Malic Acid, 0.15 M NaCl) with 0.1%Tween20 then washed four times with MAB +0.1%Tween. Samples were blocked with MAB-2% Roche Block solution for 30 min at room temperature or overnight at 4 °C. Anti-DIG AP FAB Antibody in MAB-2% Roche Block was added to the samples for a final concentration of 1:1000 and left to incubate for 2 hr at room temperature or overnight at 4 °C.

Next, the samples were washed four times in MAB-0.1%Tween20 followed by three washes in AP buffer (0.1 M NaCl, 0.1 M Tris pH 9.5), with 15 min between washes. Samples were resuspended in color reaction solution (1X AP buffer, 50 mM MgCl2, 0.2% Tween20, 0.175 mg/µL 4-Nitro blue tetrazolium chloride, 0.35 mg/mL 5-Bromo-4-chloro-3-indolyl-phosphate), transferred to watch glasses, and kept in the dark at room temperature until the reaction was completed. When the color reaction concluded, embryos were washed three times in MAB-0.1%Tween20 and stored indefinitely in MAB.

Embryos were evaluated using a Leica MZ 95 dissection scope. For imaging, embryos were resuspended in MAB- 50% glycerol, and photographs were taken using a Leica DMI 4000 B inverted scope and Leica Application Suite X V3.6.20104.

## Double fluorescent in situ hybridization

Double FISH was performed using a method previously described in *McCauley et al., 2013*. The reaction was done with a mix of 0.2 ng/µL of a digoxigenin-labelled (DIG) and 0.4 ng/µl of a dinitrophenol-labeled (DNP) antisense RNA probes. The DNP probe was detected using pre-absorbed anti-DNP-HRP antibody and labeled with Cy3-tyramide. The DIG probe was detected using a pre-absorbed anti-DIG-POD antibody and labeled with fluorescein-tyramide. Samples were imaged with a Zeiss 880 Laser Scanning Microscope with 405 nm, 488 nm, and 560 nm channels in Z-stack settings.

## Image creation and Processing

Gimp was used to process microscopy images (*The GIMP Development Team, 2021*). BioRender.com was used to create the schematics in *Figure 1A* and *Figure 8A*. Inkscape was used to compile and format the final images (*Inkscape Project, 2021*).

## Additional information

### Funding

| Funder | Grant reference number | Author |
|---|---|---|
| National Science Foundation | IOS 1557431 | Veronica Hinman |

The funders had no role in study design, data collection and interpretation, or the decision to submit the work for publication.

### Author contributions

Anne Meyer, Conceptualization, Resources, Data curation, Formal analysis, Validation, Investigation, Visualization, Methodology, Writing - original draft, Project administration, Writing - review and editing; Carolyn Ku, Conceptualization, Data curation, Formal analysis, Methodology, Writing - review and editing; William L Hatleberg, Conceptualization, Data curation, Supervision, Investigation, Methodology, Project administration, Writing - review and editing; Cheryl A Telmer, Supervision, Writing - original draft, Project administration, Writing - review and editing; Veronica Hinman, Conceptualization, Data curation, Supervision, Funding acquisition, Investigation, Methodology, Writing - original draft, Project administration, Writing - review and editing

### Author ORCIDs

Anne Meyer  http://orcid.org/0000-0003-2287-7316
William L Hatleberg  http://orcid.org/0000-0002-0423-7123
Cheryl A Telmer  http://orcid.org/0000-0002-7649-3134
Veronica Hinman  http://orcid.org/0000-0003-3414-1357

Decision letter and Author response
Decision letter https://doi.org/10.7554/eLife.80090.sa1
Author response https://doi.org/10.7554/eLife.80090.sa2

## Additional files

### Supplementary files

• Supplementary file 1. Table of LOCIDs, gene names, and gene symbols referenced in this article.

• Supplementary file 2. Table of quality control parameters used to filter the scRNA-seq and snRNA-seq datasets.

• Supplementary file 3. Table of the membership of integrated clusters. Contains the breakdown of cluster composition based on the original cell type analysis used in the single species atlases.

• Supplementary file 4. Primer sequences used to synthesize probes for WMISH.

• Supplementary file 5. In-depth annotations of cluster identity in the *S. purpuatus* snRNA-seq analysis.

• Supplementary file 6. In-depth annotations of cluster identity in *P. miniata* for clusters not discussed in the body of the paper.

• MDAR checklist

• Source code 1. An RMarkdown file containing sample code from the pipeline what was used to create the species atlases and the integrated atlases.

### Data availability

Sequencing data have been supplied as NCBI SRA indicated below. Other data is provided directly in the manuscript and supporting files.

The following datasets were generated:

| Author(s) | Year | Dataset title | Dataset URL | Database and Identifier |
| --- | --- | --- | --- | --- |
| Meyer A | 2022 | *S. purpuratus* SC sequencing | https://www.ncbi.nlm.nih.gov/bioproject/PRJNA839000 | NCBI BioProject, PRJNA839000 |
| Meyer A | 2022 | *P. miniata* SC Sequencing | https://www.ncbi.nlm.nih.gov/bioproject/PRJNA826814 | NCBI BioProject, PRJNA826814 |

The following previously published dataset was used:

| Author(s) | Year | Dataset title | Dataset URL | Database and Identifier |
| --- | --- | --- | --- | --- |
| Foster S, Wessel G, Oulhen N | 2020 | A single cell RNA-seq resource for early sea urchin development | https://www.ncbi.nlm.nih.gov/geo/query/acc.cgi?acc=GSE149221 | NCBI Gene Expression Omnibus, GSE149221 |

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
