## [Editor Report]

This important study combines single cell and single nucleus transcriptomics to two echinoderm embryos to identify embryonic cell types and assess cell type evolutionary novelties. The paper provides convincing pieces of evidence in regards to the sea star embryonic cell types and their relationship to the sea urchin ones, highlighting conserved, diverged as well as novel genetic programs operating during early echinoderm development. The work will be of broad interest to developmental and evolutionary biologists.

---

## [Decision Letter]

**Decision letter after peer review:**

Thank you for submitting your article "New hypotheses of cell type diversity and novelty from comparative single cell and nuclei transcriptomics in echinoderms" for consideration by *eLife*. Your article has been reviewed by 3 peer reviewers, including Maria Ina Arnone as the Reviewing Editor and Reviewer #1, and the evaluation has been overseen by Didier Stainier as the Senior Editor. The following individual involved in the review of your submission has agreed to reveal their identity: Roger Revilla-i-Domingo (Reviewer #2).

Essential revisions:

1) Improve validation of the P. miniata clustering by using double whole month in situ hybridization as suggested by Reviewer #1.

2) Complement the integration of sea urchin and sea star datasets analysis with a method that is not limited to the 1:1 orthologs between sea urchin and sea star, using, for example, as suggested by Reviewer #2 the algorithm based on self-assembling manyfold (SAMap). This could partially address also Reviewer #3 concern about comparing a single-nuclei RNA-seq dataset with a single-cell RNA-seq dataset.

3) Include a sea urchin gastrula stage in the sea urchin and sea star integrated single cell analysis or otherwise address the concerns of Reviewer #1 regarding the absence of a sea urchin gastrula stage in the analysis.

4) Include clear statements and general discussion about the limitation of cell type evolutionary comparison performed as described in points 2 and 3 by Reviewer #3.

*Reviewer #1 (Recommendations for the authors):*

While the manuscript is well written, particularly in the introduction and Discussion sections, the data presented show two major weaknesses, one concerning the validation of the clustering via whole month in situ hybridization (WMISH), and the other concerning the stages of S. purpuratus chosen as homologous of the ones analyzed in P. miniata.

Regarding validation by WMISH, the general point is that this reviewer would like to see more embryos/gene validation, especially for the novel marker genes identified, and, in some cases a better resolution of the mRNA localization with the help of fluorescent in situ hybridization (FISH) or double FISH. For instance, in lines 220-230, the authors make a very interesting point on novel marker genes of their neural clusters, the synaptotagmin genes, which, as shown in the dot plot reported in Figure 5a, exhibit a differential expression marking different neural cell clusters. It would be very nice to see at least a couple of FISH experiments showing these differentiating markers in situ. Double FISH for this would be perfect. Similarly, a better resolution is necessary to convince the reader about the expression patterns reported in Figure 5 E-G and those in Figure 7.

Another point regarding the WMISH is that there is no information in the paper regarding the stages analyzed. I assume that the embryos in Figure 5 and 6 correspond to the mid gastrula/ 40 h P. miniata embryos that were used as the latest time point in the construction of the single cell atlas, while the ones reported in Figure 7 correspond to later gastrula stages. However, this info has to be included in the figures and/or figure legends.

Regarding the stages of S. purpuratus chosen as homologous of the ones analyzed in P. miniata, as stated in the results and reported in Figure 3B the authors have used for P. miniate two blastula stages and one gastrula, 40h, which the authors correctly define as mid gastrula (line 171). However, strangely enough, they didn't include any gastrula stage for the corresponding sea urchin data sets, despite the fact the source they used for scRNA-seq data of S. purpuratus (Foster et al. 2020) includes two gastrula stages, one early and one late. On the contrary, the authors chose only hatched and mesenchyme blastula stages for sea urchins and justify this choice with a sentence that appears very controversial: "An extensive comparison of whole embryo RNA-seq time series data between S. purpuratus and P. miniata has shown that P. miniata develops at a faster rate and that S. purpuratus (hatched and mesenchymal blastula stages) are equivalent in developmental timing to the P.miniata time points used to generate the atlas here (Gildor et al., 2019)." Besides the fact that P. miniata definitely develops at a slower and not faster rate compared to S. purpuratus, and that the staging reported in Gildor et al. is not the best reference to be used (certainly they aren't for the two sea urchin species analysed in there) also because the 40h P. miniata stage is not included in it, this statement is clearly contradictory even in respect to what stated by the authors of this paper in other parts of the manuscript. While I can agree with the fact that the so-called "mesenchyme blastula" in sea urchins can be considered as a very early gastrula (given that mesenchyme cells are ingressed in the blastocoel at this stage), no endoderm has ingressed at all at this stage. Therefore, it would be more appropriate to include in the S. purpuratus data sets a stage in which at least some part of the archenteron has ingressed. Indeed, the S. purpuratus early gastrula of the Foster et al. dataset (30h) appears as a bona fide equivalent of the 40h P. miniata gastrula stage. I believe the exclusion of a sea urchin gastrula stage in the comparative analysis introduces a strong bias and I strongly suggest redoing this analysis with the inclusion of this stage.

Another general, important, although minor and easy to fix point, is the nomenclature used in the manuscript:

– First of all, about gene names: throughout the manuscript, text, figures, and figure legends, a plethora of different styles are used, even within the same list (such as in the y axis of dot plots), including UPPERCASE, lowercase (italics and not), capitalised words, with or without initials of species names … please choose a style and stick to it, because it is very confusing

– Second, about cluster names: some choices are in my opinion confusing as they refer to a structure/ developmental stage that is not present in the data set; this is the case for example of (left and right) "coelom", which should be better referred to as "coelomic mesoderm" given that coelomic pouches appears much later compared to the stages analysed; or the case of the "immune mesenchyme", which should be better referred to as immune-like given that immune cells, to my knowledge, were never functionally identified in the P. miniata larva; I understand that for the sake of simplicity "immune mesenchyme" could be a better name, but this assumption should be clearly specified and definitely the short name "immune cluster" used in abstract and conclusions is not appropriate

– Another issue concerns the use of "primary mesenchyme cells", which is quite an obsolete terminology, not very informative, and really meaningful only within the sea urchin community. Skeletogenic mesenchyme appears more appropriate for a wider audience.

Two other important changes that would make the manuscript more readable are:

– First, about the use of the adjusted p-value: reporting the adjusted p-value for each of the marker genes identifying each cluster appears as redundant. If a gene is a gene marker it means that it passed the adjusted p-value test and if the reader wants to know more about those figures these are all included in the tables. The result session is uselessly heavy and boring to read with all these parentheses. I strongly suggest eliminating these details in the text.

– Second, in the analysis of the contribution of cells for each integrated cluster, the list of all contributions, even if they account for just one cell, again appears redundant, and the text become difficult to read (because of all those figures, percent and parentheses) and eventually even confusing to what is the actual identity of the cluster. I would propose to decide on a certain cut-off (e.g., 10%) and list in the text only the contribution above that threshold, leaving to a supplementary figure the detailed contribution of sea urchin or sea star cluster to the integrated cluster, as done for Figure 9 and 10.

*Reviewer #2 (Recommendations for the authors):*

1. To strengthen (and potentially expand) the claims resulting from the integration of sea urchin and sea star datasets, I recommend that the authors complement their analysis with a method that is not limited to the 1:1 orthologs between sea urchin and sea star. I would suggest, for example, the algorithm based on self-assembling manyfold (SAMap), which has been successfully employed by Musser et al. (Science 2021) to compare two sponge species (Figure 2B) and has been described in detail by Tarashansky et al. (*eLife* 2021), where single-cell datasets from animals across different phyla were integrated.

2. I am not sure that this is possible, but if it is, I would suggest visualizing the data in Supplementary Table 3 in a similar way as Figure 2B of Musser et al. (Science 2021). This would make it easier to follow the description of the integrated clusters.

3. In Figure 6, the authors very nicely show that genes enriched in Pmin Cl 21 are expressed in mesenchymal populations. In particular, VEGFR1L is shown to be expressed in a population that accumulates around the base of the archenteron (Figure 6G), similar to the localization of PMCs in S. purpuratus. According to Figure 6.D, my impression is that tek (another gene enriched in Pmin CL 21) is also expressed in the population of cells around the archenteron. Although less clear, the same might be true for ets1 (Figure 6.D). Could the authors explain whether these (and perhaps other) markers of Pmin CL 21 are expressed in the mesenchymal population around the archenteron? If they are indeed expressed in this population, can the authors comment on the significance of this?

4. An interesting finding of the study by Meyer et al. is the overlap between pigment cell cluster transcriptome in sea urchins and the immune and neural clusters in sea stars. Would it be relevant to discuss this finding in the light of the hypothesis by Klimovich and Bosch (BioEssays 2018, section: "Rethinking the Role of the Nervous System") that the nervous system and the immune system have a common evolutionary origin?

*Reviewer #3 (Recommendations for the authors):*

1. Combing scRNAseq and snRNAseq data could introduce biases. Several studies have noted that seemingly subtle differences like sequencing platform or even batch can produce artificial signals within scRNAseq data, sometimes even swamping species effects (e.g., Mizrahi-Man and Gilad 2015 PMID: 26236466; Tung et al. 2017 PMID: 28045081). Thus, it's not clear to me that single-nucleus and single-cell RNAseq data can be combined without introducing systematic biases in the representation of transcripts present. For instance, mRNAs with short half-lives would likely be over-represented in snRNAseq compared with scRNAseq. I suspect that conclusions about overlapping clusters and the presence/absence of clusters reported in this study are generally secure. However, differences in timing and in proportions of cells in a given cluster between species may be more problematic. If the authors are aware of any empirical data that support direct comparisons between snRNAseq and scRNA seq, citing these would build confidence in the results. At a minimum, the authors should state more clearly that they are making an assumption when combining different data types in this study, and also comment on how this might affect the interpretation of results.

2. Sampling only embryonic stages limits the ability to confidently identify the absence of a cell type. In an adult organ composed mostly of differentiated cells, determining the presence/absence of a cell type is fairly straightforward, assuming fairly representative cells/nuclei and reasonably high-quality data. In contrast, a gastrula-stage embryo contains very few, if indeed any, fully differentiated cells. Further, it is well established that homologous cells can differentiate at different stages of development in different species. Thus, a cell type might appear to be absent in the gastrula of one species and present in another simply because it has not yet begun to differentiate in the first species. It's a situation where the absence of evidence is not evidence of absence. This makes it impossible to reach any firm conclusions about which cell types might be unique to seastars or sea urchins. The authors are clearly aware of this limitation, e.g. lines 446-447: "thus making Int CL 16 the only S. purpuratus-specific cell type at this stage of development" (emphasis added). Here and elsewhere, the authors are good about stating this caveat. However, these statements come across as each applying to a specific case rather than a general limitation that applies to all cell types. This more general caveat should be clearly stated, as it is key to understanding what can and can't be learned from the data. This is particularly important because many non-specialists reading this paper will get lost in, or simply skip over, the numerous details presented for any given cell type and could easily miss both the individual caveats and the more general point that in no case is it possible to conclude that a particular cell type is present or absent from one species or the other.

3. Data from only two species limits the ability to confidently identify the direction of evolutionary change. Even if we accept that some cells are truly present/absent, figuring out whether these cases represent novelties or losses is problematic when only examining two species: is it a gain in the species with the unique cell type or a loss in the species without? The usual approach to answering this question is getting data from an outgroup species, but in the present case, the data simply aren't available. The authors are not always clear about this limitation, for example (lines 462-463) "this cluster can be considered novel to S. purpuratus". More accurate would be to say that it is uniquely present rather than novel. It could also be the case that the ancestral echinoderm had an early specification of germ cells and that this has been lost in seastars and perhaps other echinoderms. Here again, the limitation is a general one rather than specific to this particular cell type and should be clearly stated as such. Again, this is key to understanding what can and cannot be inferred from the data.

A suggestion: There may be an opportunity to make a different kind of evolutionarily inference that is more secure, namely to identify evolutionary differences in the timing of differentiation of specific cell types. These observations are interesting because they have implications for the evolution of cell fate specification mechanisms. For instance, germ cells are specified clonally and very early in sea urchins and the molecular mechanisms and cellular processes responsible are well studied. The absence of any cells with a similar transcriptional profile in seastar embryos implies no that early segregation of the germ line does not occur, and thus it is unlikely that localized maternal determinants are responsible. Given the wealth of information available about cell fate specification mechanisms in sea urchins, inferences about evolutionary differences could be made about other cell types as well. This seems to me a missed opportunity for the authors to highlight some interesting and informative aspects of their results.

---

## [Author Response]

Essential revisions:1) Improve validation of the P. miniata clustering by using double whole month in situ hybridization as suggested by Reviewer #1.

We have added single-color WMISH for the genes *syt14* (Figure 5.i), *syt7* (Figure 5.h), *etv1* (Figure 5.f) and *got1* (Figure 5.g). These genes were identified as markers for the four neural clusters and now importantly indicate expression in patterns consistent with neural fate and a pattern that differs between the genes suggesting non overlapping of neural fates. We additionally added two color FISH for *gcml* + *hoxd9a* (Figure 5.l) and *rab3* + *gcml* (Figure 5.M). These were important to show that *hoxd9a* and *rab3* are coexpressed with some subsets of *gcml*+ cells. This supports the hypothesis suggested by the clustering of neural types in Figure 4 and Figure 5.A that *gcml*, *rab3* and *hoxd9a* are are marker genes for the Sensory Neural cluster (Pm CL 22) and colocalize. There is not perfect overlap with these gene transcripts suggesting an even more complex neural cell fate than we have captured in our clustering. We also performed FISH for *ets1* + *tek* (Figure 6.h) to support the hypothesis of Figure 4 and Figure 6.A and 6.B that *ets1* and *tek* are coexpressed and are marker genes for immune-like mesenchyme (Pm CL 1). We show that they are exclusively overlapping.

We have added text into the results, figure legends, and discussion to provide details about these experiments and their interpretation.

2) Complement the integration of sea urchin and sea star datasets analysis with a method that is not limited to the 1:1 orthologs between sea urchin and sea star, using, for example, as suggested by Reviewer #2 the algorithm based on self-assembling manyfold (SAMap). This could partially address also Reviewer #3 concern about comparing a single-nuclei RNA-seq dataset with a single-cell RNA-seq dataset.

After careful consideration we have decided to maintain our focus on 1:1 orthologs for our mapping as we believe this gives more robust results SAMap was developed for mouse/human integration (~100 Mya) and will be limited for sea star/sea urchin integration (~400 Mya). First it should be noted that our orthology mapping between these two species has been extensively performed using the principles of the Alliance of Genome Resources that at least three DIOPT tools should agree on the orthology. We have used BLAST, phylogenetic-based tools (InParanoid v4.1, ProteinOrtho v6, SwiftOrtho, FastOrtho, OMA and OrthoFinder) as published in “Integration of 1: 1 orthology maps and updated datasets into Echinobase, Foley et al. Database, 2021”. We think mapping using orthologous genes provides a more reliable comparison for our integrated echinoderm datasets.

We also show that several results from this integrated mapping corroborate observations from other data types to support that we are making meaningful findings. For example, we identify a sea urchin specific integrated cluster (Int CL 14) that corresponds to *S. purpuratus* Piwil3+ Cells. The near absence of sea star nuclei in Int CL 14 is supported by previous gene expression studies (Fresques et al. 2014) which show that *piwil3*, *nanos2l* and *pum2* orthologs/paralogs are not expressed in these stages in *P. miniata*.

To reflect our focus on ortholog driven clustering we have changed the title to of the paper

New hypotheses of cell type diversity and novelty from orthology-driven comparative single cell and nuclei transcriptomics in echinoderms.

We also comment here on the differences between single nuclei and single cell approaches. We carefully considered the approaches to isolation and thought for this and future studies that single nuclei sequencing would be less biased, robust and easier to achieve reliability; especially as researchers look to later and more complex stages and tissues. We therefore took great care to evaluate the two procedures since we do rely on comparison to scRNA sequencing. Figure 2 and Figure 2 Supplemental FIgure 1 strongly support that there are no significant differences between approaches and that they can be reliably compared. We showed bulk RNA profiles are not significantly different when obtained from whole embryos, isolated cells or isolated nuclei (Figure 2) and that the single nuclei atlas prepared in our studies was highly similar to the atlas previously prepared from single cells (Foster et al. 2019).

We added the following sentence in the fourth paragraph of the results to more thoroughly clarify our position on this.

"These results thus strongly support that single nuclei and single-cell RNA isolation and sequencing each resulted in appropriate representative clusters thus providing confidence that either isolation method was an effective way to establish cell clusters. snRNA-seq is more likely to capture earlier transcriptional responses than scRNA-seq, but such timing differences are much smaller than the sampling times here and unlikely to affect our analyses."

3) Include a sea urchin gastrula stage in the sea urchin and sea star integrated single cell analysis or otherwise address the concerns of Reviewer #1 regarding the absence of a sea urchin gastrula stage in the analysis.

We have included this new time point in our analysis. In doing so we have re-clustered and re-annotated the sea urchin data and re-performed the integrated cluster analysis. This reanalysis has led to many small changes in summary statistics and specifics of clusters, and overlapping gene sets. However, the main findings of the work and conclusions of the analyses have not changed, as many of the differences in the re-analyses are of small effect.

Thus, we recreated the sea urchin developmental atlas now using three time points (hatched blastula, and mesenchymal blastula, and early gastrula stages). We now have 10,611 cells in the analysis rather than the original reported 7,370 and 23 vs 20 distinct clusters.

We then performed a new cluster annotation. The most significant difference with this new analysis is that we now have two rather than one pigment cell cluster, named Spur CL 12 Pigment Cells A and Spur CL 14 Pigment Cells B. Additionally small micromeres and muscle clusters were annotated. The *S. purpuratus* annotation in Figure 8 supplemental figure 1 and Figure 8 was updated to reflect the new cluster names.

We added additional text to more robustly explain the annotation of Spur CL 12 and 14, the two pigment cell clusters and Spur CL 13 as skeletogenic mesenchyme (previously PMCs).

The text now reads:

“Two clusters were identified as belonging to the pigment cell lineage (Spur CL 12 Pigment Cells A and Spur CL 14 Pigment Cells B). These clusters shared 107 markers including known markers of pigment cells in sea urchins, including *gcml*, *polyketide synthase 1 (pks1), dimethylaniline monooxygenase [N-oxide-forming] 2 (fmo3),* and *sulfotransferase* (*sult*) (Calestani et al. 2003). We hypothesize Spur CL 14 Pigment cells B represents a population of pigment cells in an earlier state of differentiation than Spur CL 12 Pigment Cells A, because upstream transcription factors such as *etv6, gata6, and etv1* mark only Spur CL 14 Pigment Cells B. “

“The skeletogenic mesenchyme cluster (Spur CL 13) was marked by the expression of regulatory genes such as ets1, aristaless-like homeobox (alx1), ALX homeobox 1-like (alx1l, previously called alx4), T-box brain transcription factor 1 (tbr), ets variant transcription factor 6 (etv6, previously called tel), transcriptional regulator ERG (erg), hematopoietically-expressed homeobox protein HHEX homolog (hhex), TGFB induced factor homeobox 2-like (tgif2), arid3a, and vascular endothelial growth factor receptor 1 (flt1, also known as vegfr1). As expected, this cluster was also marked by genes involved in biomineralization, including 27 kDa primary mesenchyme-specific spicule protein (pm27a), spicule matrix protein SM50 (sm50), and mesenchyme-specific cell surface glycoprotein (msp130).”

While the annotation of the immune cell cluster did not change, we added additional text to more clearly explain the rationale for this cluster.

"Another cluster of particular interest is Spur CL 17 Oral Non-skeletogenic Mesenchyme, the lineage fated to become immune coelomocyte cells. This cluster is marked by genes known to be involved in the specification of this lineage like *erg, gata3, gata6, hhex, prox1, ets1,* as well as genes that have been linked to immune response in echinoderms, like *tek*, and *srcr*. Other genes implicated in immune function in other systems such as two allograft inflammatory factor 1-like genes (LOC763226, LOC115929190), C-type lectin domain-containing protein 141-like, and duox1 are also found in this cluster (Brown et al., 2018; Rada and Leto, 2008; Vizioli et al., 2020)."

With the new cluster analysis we then re-performed the integration between the *P. miniata* and *S. purpuratus* datasets and updated the analyses in Figures 8-12.

We have used the same methodology and pipelines as previously. As requested by Reviewer 3 we have provided the code used to create and integrate the atlases in Supplementary file 7.

The integrated analysis with the additional data changed some of the values and numbers of overlapping genes but no substantive differences were found in our conclusions. We identified 15 rather than 17 clusters in this new dataset.

The wording has been changed in multiple places in the Results section to reflect the changed numbering of the clusters (simply a result of reclustering) and the % and numbers of overlapping genes and changes in significance. Summarized below, these differences were only slight and did not change the conclusions.

Figure 9 shows that Int CL 14 is enriched with cells from Spur CL 22 piwil3+ Cells and Undetermined with very few Pmin CL 19 Dividing Ectoderm.

Figure 10 shows the integrated clusters with separate contributions from Pigment A and B. Int CL 8 and 12 show contributions from both Pigment A and B (Spur). Contributions from Pmin to Int CL 8 are majority Immune-like Mesenchyme and Int CL 12 shows contributions from early, mixed and sensory neural clusters from *P. miniata* but now with stronger support (RF 5.2 compared to previous 5.0) although the number of genes overlapping decreased somewhat.

Figure 11 shows that Int CL 7 is primarily *S. purpuratus* Skeletogenic Mesenchyme and P. miniata Right Coelom and Foregut. Our conclusions here therefore remain unchanged also.

Figure 12 shows Int CL 10 is primarily composed of *P. miniata* Left Coelom with no predominant corresponding contributions from *S. purpuratus.* This is the same result also.

4) Include clear statements and general discussion about the limitation of cell type evolutionary comparison performed as described in points 2 and 3 by Reviewer #3.

We agree that these reviewers have made important points and we have added several new sentences throughout the text and in particular into the conclusion section to address their comments.

We have added into the discussion:

“This suggests that these genes may play dual roles in immune and neural cells and the emergence of modern pigment cells is the result of a fusion of regulatory programs from different cell lineages. Future work comparing fully differentiated neurons, immune cells and pigment cells would help elucidate any further similarities in cell function and gene expression, possibly providing insights into how novel cell types arise and develop unique phenotypes via the co-option of regulatory pathways.”

And into the conclusions

“The examples uncovered in this study serve to expand our understanding of evolution of cell types among these echinoderms, and more generally highlight ways that we can think about novelty and the limitations of experimental approaches to this understanding.”

“Novelty in both of these cases may be relative changes in timing or heterochrony of the cell state specification. Unless snRNA-seq covers a tight time course across the entirety of development and adult tissues, and possible environmental contexts we can never truly determine novelty. However heterochronic activation of a GRN in a new context can result in dramatic functional differences to the organism. The very early specification of a *piwil3+* cell population in sea urchin likely points to a different developmental program and possible functional role of these cell types in these species.”

Reviewer #1 (Recommendations for the authors):While the manuscript is well written, particularly in the introduction and Discussion sections, the data presented show two major weaknesses, one concerning the validation of the clustering via whole month in situ hybridization (WMISH), and the other concerning the stages of S. purpuratus chosen as homologous of the ones analyzed in P. miniata.Regarding validation by WMISH, the general point is that this reviewer would like to see more embryos/gene validation, especially for the novel marker genes identified, and, in some cases a better resolution of the mRNA localization with the help of fluorescent in situ hybridization (FISH) or double FISH. For instance, in lines 220-230, the authors make a very interesting point on novel marker genes of their neural clusters, the synaptotagmin genes, which, as shown in the dot plot reported in Figure 5a, exhibit a differential expression marking different neural cell clusters. It would be very nice to see at least a couple of FISH experiments showing these differentiating markers in situ. Double FISH for this would be perfect. Similarly, a better resolution is necessary to convince the reader about the expression patterns reported in Figure 5 E-G and those in Figure 7.

These in situs have been added and details are provided under “essential revisions” point 1.

We have added single-color WMISH of 48 hpf gastrula stage embryos for the genes *syt14* (Figure 5.I), *syt7* (Figure 5.H), *etv1* (Figure 5.F) and *got1* (Figure 5.G). These genes were identified as markers for the four neural clusters and now importantly indicate expression in patterns consistent with neural fate and a pattern that differs between the genes suggesting non overlapping of neural fates. We additionally added two color FISH for *gcml* + *hoxd9a* (Figure 5.l) and *rab3* + *gcml* (Figure 5.M). These were important to show that *hoxd9a* and *rab3* are coexpressed with some subsets of *gcml*+ cells. This supports the hypothesis suggested by the clustering of neural types in Figure 4 and Figure 5.A that *gcml*, *rab3* and *hoxd9a* are are marker genes for the Sensory Neural cluster (Pmin CL 22) with some colocalization. There is not perfect overlap with these gene transcripts suggesting an even more complex neural cell fate than we have captured in our clustering.

We also added images from double FISH of 48 hpf gastrula stage embryos for *ets1* + *tek* (Figure 6.H) to support the hypothesis of Figure 4 and Figure 6.A and 6.B that *ets1* and *tek* are coexpressed and are marker genes for Immune-like Mesenchyme (Pm CL 1). We show that they are exclusively overlapping.

Figure 7.D-I are WMISH of later stages of development (72 hpf). The 48 hpf gastrula stage embryos show no morphological distinction between left and right so we performed in situs at a later stage where the left/right symmetry exists to show the expression in the coeloms. WMISH served to show this asymmetrical expression.

We have added text into the results, figure legends, and discussion to provide details about these experiments and their interpretation.

Another point regarding the WMISH is that there is no information in the paper regarding the stages analyzed. I assume that the embryos in Figure 5 and 6 correspond to the mid gastrula/ 40 h P. miniata embryos that were used as the latest time point in the construction of the single cell atlas, while the ones reported in Figure 7 correspond to later gastrula stages. However, this info has to be included in the figures and/or figure legends.

Times and stages have been added to the figure legends.

Figure 5 shows 48 hpf gastrula stage embryos.

Figure 6 shows 48 hpf gastrula stage embryos.

Figure 7 shows 72 hpf gastrula stage embryos.

Regarding the stages of S. purpuratus chosen as homologous of the ones analyzed in P. miniata, as stated in the results and reported in Figure 3B the authors have used for P. miniate two blastula stages and one gastrula, 40h, which the authors correctly define as mid gastrula (line 171). However, strangely enough, they didn't include any gastrula stage for the corresponding sea urchin data sets, despite the fact the source they used for scRNA-seq data of S. purpuratus (Foster et al. 2020) includes two gastrula stages, one early and one late. On the contrary, the authors chose only hatched and mesenchyme blastula stages for sea urchins and justify this choice with a sentence that appears very controversial: "An extensive comparison of whole embryo RNA-seq time series data between S. purpuratus and P. miniata has shown that P. miniata develops at a faster rate and that S. purpuratus (hatched and mesenchymal blastula stages) are equivalent in developmental timing to the P.miniata time points used to generate the atlas here (Gildor et al., 2019)." Besides the fact that P. miniata definitely develops at a slower and not faster rate compared to S. purpuratus, and that the staging reported in Gildor et al. is not the best reference to be used (certainly they aren't for the two sea urchin species analysed in there) also because the 40h P. miniata stage is not included in it, this statement is clearly contradictory even in respect to what stated by the authors of this paper in other parts of the manuscript. While I can agree with the fact that the so-called "mesenchyme blastula" in sea urchins can be considered as a very early gastrula (given that mesenchyme cells are ingressed in the blastocoel at this stage), no endoderm has ingressed at all at this stage. Therefore, it would be more appropriate to include in the S. purpuratus data sets a stage in which at least some part of the archenteron has ingressed. Indeed, the S. purpuratus early gastrula of the Foster et al. dataset (30h) appears as a bona fide equivalent of the 40h P. miniata gastrula stage. I believe the exclusion of a sea urchin gastrula stage in the comparative analysis introduces a strong bias and I strongly suggest redoing this analysis with the inclusion of this stage.

We have now added the sea urchin early gastrula (24 hpf) stage sample from Foster et al. to the data and these changes to the manuscript are explained in “essential revisions” point 3.

We have included this new time point in our analysis. In doing so we have re-clustered and re-annotated the sea urchin data and re-performed the integrated cluster analysis. This reanalysis has led to many small changes in summary statistics and specifics of clusters, and overlapping gene sets. However, the main findings of the work and conclusions of the analyses have not changed, as many of the differences in the re-analyses are of small effect.

Thus, we recreated the sea urchin developmental atlas now using three time points (hatched blastula, and mesenchymal blastula, and early gastrula stages). We now have 10,611 cells in the analysis rather than the original reported 7,370 and 23 vs 20 distinct clusters.

We then performed a new cluster annotation. The most significant difference with this new analysis is that we now have two rather than one pigment cell cluster, named Spur CL 12 Pigment Cells A and Spur CL 14 Pigment Cells B. Additionally small micromeres and muscle clusters were annotated. The *S. purpuratus* annotation in Figure 8 Supplemental Figure 1 and Figure 8 was updated to reflect the new cluster names.

We added additional text to more robustly explain the annotation of Spur CL 12 and 14, the two pigment cell clusters and Spur CL 13 as skeletogenic mesenchyme (previously PMCs).

The text now reads:

“Two clusters were identified as belonging to the pigment cell lineage (Spur CL 12 Pigment Cells A and Spur CL 14 Pigment Cells B). These clusters shared 107 markers including known markers of pigment cells in sea urchins, including gcml, polyketide synthase 1 (pks1), dimethylaniline monooxygenase [N-oxide-forming] 2 (fmo3), and sulfotransferase (sult) (Calestani et al. 2003). We hypothesize Spur CL 14 Pigment cells B represents a population of pigment cells in an earlier state of differentiation than Spur CL 12 Pigment Cells A, because upstream transcription factors such as *etv6, gata6, and etv1* mark only Spur CL 14 Pigment Cells B. “

“The skeletogenic mesenchyme cluster (Spur CL 13) was marked by the expression of regulatory genes such as ets1, aristaless-like homeobox (alx1), ALX homeobox 1-like (alx1l, previously called alx4), T-box brain transcription factor 1 (tbr), ets variant transcription factor 6 (etv6, previously called tel), transcriptional regulator ERG (erg), hematopoietically-expressed homeobox protein HHEX homolog (hhex), TGFB induced factor homeobox 2-like (tgif2), arid3a, and vascular endothelial growth factor receptor 1 (flt1, also known as vegfr1). As expected, this cluster was also marked by genes involved in biomineralization, including 27 kDa primary mesenchyme-specific spicule protein (pm27 a), spicule matrix protein SM50 (sm50), and mesenchyme-specific cell surface glycoprotein (msp130).”

While the annotation of the immune cell cluster did not change, we added additional text to more clearly explain the rationale for this cluster.

"Another cluster of particular interest is Spur CL 17 Oral Non-skeletogenic mesenchyme, the lineage fated to become immune coelomocyte cells. This cluster is marked by genes known to be involved in the specification of this lineage like *erg, gata3, gata6, hhex, prox1, ets1,* as well as genes that have been linked to immune response in echinoderms, like *tek*, and *srcr*. Other genes implicated in immune function in other systems such as two allograft inflammatory factor 1-like genes (LOC763226, LOC115929190), C-type lectin domain-containing protein 141-like, and duox1 are also found in this cluster (Brown et al., 2018; Rada and Leto, 2008; Vizioli et al., 2020)."

With the new cluster analysis we then re-performed the integration between the *P. miniata* and *S. purpuratus* datasets and updated the analyses in Figures 8-12.

We have used the same methodology and pipelines as previously. As requested by Reviewer 3 we have provided the code used to create and integrate the atlases in Supplementary file 7.

The integrated analysis with the additional data changed some of the values and numbers of overlapping genes but no substantive differences were found in our conclusions. We identified 15 rather than 17 clusters in this new dataset.

The wording has been changed in multiple places in the Results section to reflect the changed numbering of the clusters (simply a result of reclustering) and the % and numbers of overlapping genes and changes in significance. Summarized below, these differences were only slight and did not change the conclusions.

Figure 9 shows that Int CL 14 is enriched with cells from Spur CL 22 Piwil3+ cells and Undetermined with very few Pmin CL 19 Dividing Ectoderm.

Figure 10 shows the integrated clusters with separate contributions from Pigment A and B. Int CL 8 and 12 show contributions from both Pigment A and B (Spur). Contributions from Pmin to Int CL 8 are majority Immune-like Mesenchyme and Int CL 12 shows contributions from early, mixed and sensory neural clusters from *P. miniata* but now with stronger support (RF 5.2 compared to previous 5.0) although the number of genes overlapping decreased somewhat.

Figure 11 shows that Int CL 7 is primarily *S. purpuratus* Skeletogenic Mesenchyme and P. miniata Right Coelom and Foregut. Our conclusions here therefore remain unchanged also.

Figure 12 shows Int CL 10 is primarily composed of *P. miniata* Left Coelom with no predominant corresponding contributions from *S. purpuratus.* This is the same result also.

Another general, important, although minor and easy to fix point, is the nomenclature used in the manuscript:– First of all, about gene names: throughout the manuscript, text, figures, and figure legends, a plethora of different styles are used, even within the same list (such as in the y axis of dot plots), including UPPERCASE, lowercase (italics and not), capitalised words, with or without initials of species names … please choose a style and stick to it, because it is very confusing

We have followed the Echinobase Gene Nomenclature Guidelines where possible (https://www.echinobase.org/entry/static/gene/geneNomenclature.jsp). Briefly, gene symbols and names will be lowercase and italicized and follow HGNC guidelines. Echinobase currently has *P. miniata* gene symbols for 1:1 orthologs to *S. purpuratus* therefore the NCBI-annotated gene symbols or locus IDs (LOCentrezid#) are provided. Supplementary Table 1 lists gene symbols and names from Echinobase, NCBI and Ensembl for all genes mentioned in this manuscript.

– Second, about cluster names: some choices are in my opinion confusing as they refer to a structure/ developmental stage that is not present in the data set; this is the case for example of (left and right) "coelom", which should be better referred to as "coelomic mesoderm" given that coelomic pouches appears much later compared to the stages analysed; or the case of the "immune mesenchyme", which should be better referred to as immune-like given that immune cells, to my knowledge, were never functionally identified in the P. miniata larva; I understand that for the sake of simplicity "immune mesenchyme" could be a better name, but this assumption should be clearly specified and definitely the short name "immune cluster" used in abstract and conclusions is not appropriate

We agree that this can be misleading, but also we want to convey our best understanding of the functional types in these clusters. We have throughput therefore softened naming for example from immune to immune-like in the *P. miniata* atlas and PGCs to Piwil3+ cells in the *S. purpuratus* atlas.

Added text:

“Spur CL 22 is annotated as piwil3+ Cells, because they are marked by the expression of *piwi-like RNA-mediated gene silencing 3* (*piwil3*). It is also marked by the gene nanos *C2HC-type zinc finger 2-like (nanos2l),* which has been shown to play a role in Primordial Germ Cell (PGC) specification (Juliano et al. 2010, Yajima et al. 2014). However, it must be noted that while this cluster could potentially be PGCs, further studies are needed to determine if this population is simply a *piwil3* positive group of cells with stem cell like properties or a genuine germ cell lineage.”

“Pmin CL 21 is a very spatially distinct cluster in the UMAP dimensional reduction plot (Figure 6.A) and was identified as Immune-like Mesenchyme.*”*

For the coelom clusters we are confident in the left/right symmetry based on the WMISH data of the cluster markers shown in Figure 7 of a 72 hpf late gastrula stage where the left and right coeloms are clearly distinguishable.

– Another issue concerns the use of "primary mesenchyme cells", which is quite an obsolete terminology, not very informative, and really meaningful only within the sea urchin community. Skeletogenic mesenchyme appears more appropriate for a wider audience.

Thank you for this comment, we have changed primary mesenchyme cells (PMCs) to skeletogenic mesenchyme throughout the manuscript.

Two other important changes that would make the manuscript more readable are:– First, about the use of the adjusted p-value: reporting the adjusted p-value for each of the marker genes identifying each cluster appears as redundant. If a gene is a gene marker it means that it passed the adjusted p-value test and if the reader wants to know more about those figures these are all included in the tables. The result session is uselessly heavy and boring to read with all these parentheses. I strongly suggest eliminating these details in the text.

We have removed these values throughout the text.

– Second, in the analysis of the contribution of cells for each integrated cluster, the list of all contributions, even if they account for just one cell, again appears redundant, and the text become difficult to read (because of all those figures, percent and parentheses) and eventually even confusing to what is the actual identity of the cluster. I would propose to decide on a certain cut-off (e.g., 10%) and list in the text only the contribution above that threshold, leaving to a supplementary figure the detailed contribution of sea urchin or sea star cluster to the integrated cluster, as done for Figure 9 and 10.

We have removed these throughout the text.

Reviewer #2 (Recommendations for the authors):1. To strengthen (and potentially expand) the claims resulting from the integration of sea urchin and sea star datasets, I recommend that the authors complement their analysis with a method that is not limited to the 1:1 orthologs between sea urchin and sea star. I would suggest, for example, the algorithm based on self-assembling manyfold (SAMap), which has been successfully employed by Musser et al. (Science 2021) to compare two sponge species (Figure 2B) and has been described in detail by Tarashansky et al. (eLife 2021), where single-cell datasets from animals across different phyla were integrated.

We discuss this under “essential revisions” point 2 where we justify our decision to maintain the analysis on 1:1 orthologs and acknowledge this by changing the title of the paper to reflect our focus on this subset of genes.

2. I am not sure that this is possible, but if it is, I would suggest visualizing the data in Supplementary Table 3 in a similar way as Figure 2B of Musser et al. (Science 2021). This would make it easier to follow the description of the integrated clusters.

While we appreciate this reviewer's point after discussion among the authors, we opted for our visualization method used in Figures 9 -12 because we determined a ribbon plot linking 3 clustering methods was ultimately too complex and chaotic to effectively communicate our findings. Therefore, we opted to use bar plots to examine in depth the integrated clusters of interest while providing all the cluster composition information in a supplementary table that is more easily navigated.

3. In Figure 6, the authors very nicely show that genes enriched in Pmin Cl 21 are expressed in mesenchymal populations. In particular, VEGFR1L is shown to be expressed in a population that accumulates around the base of the archenteron (Figure 6G), similar to the localization of PMCs in S. purpuratus. According to Figure 6.D, my impression is that tek (another gene enriched in Pmin CL 21) is also expressed in the population of cells around the archenteron. Although less clear, the same might be true for ets1 (Figure 6.D). Could the authors explain whether these (and perhaps other) markers of Pmin CL 21 are expressed in the mesenchymal population around the archenteron? If they are indeed expressed in this population, can the authors comment on the significance of this?

We have added more in situs and double in situs and have added text as now explained in “essential revisions” point 1.

We have also added the following text:

"This is an intriguing finding as it suggests that the patterning and migration cues that direct skeletogenic mesoderm cells in sea urchins are also in place in sea stars. This indicates that perhaps only the very final activation of the skeletogenesis program is present in sea urchins and that otherwise the cell lineage between these species is similar. This finding is also consistent with the findings from Cidaroid sea urchins that show that skeletogenic cells arise from a population of mesenchyme that migrates from the top of the archenteron (Erkenbrack and Davidson 2015)

4. An interesting finding of the study by Meyer et al. is the overlap between pigment cell cluster transcriptome in sea urchins and the immune and neural clusters in sea stars. Would it be relevant to discuss this finding in the light of the hypothesis by Klimovich and Bosch (BioEssays 2018, section: "Rethinking the Role of the Nervous System") that the nervous system and the immune system have a common evolutionary origin?

We agree that this is a very interesting finding. We have added the following text and reference into the discussion:

"This supports previous work hypothesizing the close evolutionary relationship and possible functional ancestral interdependence between the immune and nervous systems (Klimovich and Bosch, 2018; Kraus et al., 2021)."

Reviewer #3 (Recommendations for the authors):1. Combing scRNAseq and snRNAseq data could introduce biases. Several studies have noted that seemingly subtle differences like sequencing platform or even batch can produce artificial signals within scRNAseq data, sometimes even swamping species effects (e.g., Mizrahi-Man and Gilad 2015 PMID: 26236466; Tung et al. 2017 PMID: 28045081). Thus, it's not clear to me that single-nucleus and single-cell RNAseq data can be combined without introducing systematic biases in the representation of transcripts present. For instance, mRNAs with short half-lives would likely be over-represented in snRNAseq compared with scRNAseq. I suspect that conclusions about overlapping clusters and the presence/absence of clusters reported in this study are generally secure. However, differences in timing and in proportions of cells in a given cluster between species may be more problematic. If the authors are aware of any empirical data that support direct comparisons between snRNAseq and scRNA seq, citing these would build confidence in the results. At a minimum, the authors should state more clearly that they are making an assumption when combining different data types in this study, and also comment on how this might affect the interpretation of results.

We have provided comments on this in Essential Revisions under point 2.

After careful consideration we have decided to maintain our focus on 1:1 orthologs for our mapping as we believe this gives more robust results SAMap was developed for mouse/human integration (~100 Mya) and will be limited for sea star/sea urchin integration (~400 Mya). First it should be noted that our orthology mapping between these two species has been extensively performed using the principles of the Alliance of Genome Resources that at least three DIOPT tools should agree on the orthology. We have used BLAST, phylogenetic-based tools (InParanoid v4.1, ProteinOrtho v6, SwiftOrtho, FastOrtho, OMA and OrthoFinder) as published in “Integration of 1: 1 orthology maps and updated datasets into Echinobase, Foley et al. Database, 2021”. We think mapping using orthologous genes provides a more reliable comparison for our integrated echinoderm datasets.

We also show that several results from this integrated mapping corroborate observations from other data types to support that we are making meaningful findings. For example, we identify a sea urchin specific integrated cluster (Int CL 14) that corresponds to *S. purpuratus* Piwil3+ Cells. The near absence of sea star nuclei in Int CL 14 is supported by previous gene expression studies (Fresques et al. 2014) which show that *piwil3*, *nanos2l* and *pum2* orthologs/paralogs are not expressed in these stages in *P. miniata*.

To reflect our focus on ortholog driven clustering we have changed the title to of the paper.

New hypotheses of cell type diversity and novelty from orthology-driven comparative single cell and nuclei transcriptomics in echinoderms.

We also comment here on the differences between single nuclei and single cell approaches. We carefully considered the approaches to isolation and thought for this and future studies that single nuclei sequencing would be less biased, robust and easier to achieve reliability; especially as researchers look to later and more complex stages and tissues. We therefore took great care to evaluate the two procedures since we do rely on comparison to scRNA sequencing. Figure 2 and Supplementary file 5 strongly support that there are no significant differences between approaches and that they can be reliably compared. We showed bulk RNA profiles are not significantly different when obtained from whole embryos, isolated cells or isolated nuclei (Figure 2) and that the single nuclei atlas prepared in our studies was highly similar to the atlas previously prepared from single cells (Foster et al. 2019).

We added the following sentence in the fourth paragraph of the results to more thoroughly clarify our position on this.

"These results thus strongly support that single nuclei and single-cell RNA isolation and sequencing each resulted in appropriate representative clusters thus providing confidence that either isolation method was an effective way to establish cell clusters. snRNA-seq is more likely to capture earlier transcriptional responses than scRNA-seq, but such timing differences are much smaller than the sampling times here and unlikely to affect our analyses."

2. Sampling only embryonic stages limits the ability to confidently identify the absence of a cell type. In an adult organ composed mostly of differentiated cells, determining the presence/absence of a cell type is fairly straightforward, assuming fairly representative cells/nuclei and reasonably high-quality data. In contrast, a gastrula-stage embryo contains very few, if indeed any, fully differentiated cells. Further, it is well established that homologous cells can differentiate at different stages of development in different species. Thus, a cell type might appear to be absent in the gastrula of one species and present in another simply because it has not yet begun to differentiate in the first species. It's a situation where the absence of evidence is not evidence of absence. This makes it impossible to reach any firm conclusions about which cell types might be unique to seastars or sea urchins. The authors are clearly aware of this limitation, e.g. lines 446-447: "thus making Int CL 16 the only S. purpuratus-specific cell type at this stage of development" (emphasis added). Here and elsewhere, the authors are good about stating this caveat. However, these statements come across as each applying to a specific case rather than a general limitation that applies to all cell types. This more general caveat should be clearly stated, as it is key to understanding what can and can't be learned from the data. This is particularly important because many non-specialists reading this paper will get lost in, or simply skip over, the numerous details presented for any given cell type and could easily miss both the individual caveats and the more general point that in no case is it possible to conclude that a particular cell type is present or absent from one species or the other.

We agree with the reviewer that it can be extremely difficult to demonstrate that a cell type is entirely absent in one species and hence truly novel in the other. We think the novelty we can assess here is developmental timing and context. We have tried to be more clear about the role of timing differences in what we have found, which was also suggested by reviewer 2.

Text in intro:

“Their larval forms in particular have several cell types that, based on developmental, morphological, and functional criteria, are considered to be novel (Figure 1)."

Text added to discussion:

“Further work examining these cell types across species of echinoderms is needed to better define these novelties and allow for a more comprehensive characterization of their evolutionary timing.”

“Additionally, work comparing fully differentiated neurons, immune cells and pigment cells would help elucidate any further similarities in cell function and gene expression, possibly providing insights into how novel cell types arise and develop unique phenotypes via the co-option of regulatory pathways.”

3. Data from only two species limits the ability to confidently identify the direction of evolutionary change. Even if we accept that some cells are truly present/absent, figuring out whether these cases represent novelties or losses is problematic when only examining two species: is it a gain in the species with the unique cell type or a loss in the species without? The usual approach to answering this question is getting data from an outgroup species, but in the present case, the data simply aren't available. The authors are not always clear about this limitation, for example (lines 462-463) "this cluster can be considered novel to S. purpuratus". More accurate would be to say that it is uniquely present rather than novel. It could also be the case that the ancestral echinoderm had an early specification of germ cells and that this has been lost in seastars and perhaps other echinoderms. Here again, the limitation is a general one rather than specific to this particular cell type and should be clearly stated as such. Again, this is key to understanding what can and cannot be inferred from the data.

We agree with this reviewer and have worked to be more clear and not assume direction and be more careful about what we have discussed as novel versus simply missing at this time point. We have addressed this point also in the essential revisions.

We have added the following text:

“Therefore, this cluster can be considered novel to *S. purpuratus* within our sampled window.”

"Further work examining these cell types across species of echinoderms is needed to better define these novelties and allow for a more comprehensive characterization of their evolutionary timing."

A suggestion: There may be an opportunity to make a different kind of evolutionarily inference that is more secure, namely to identify evolutionary differences in the timing of differentiation of specific cell types. These observations are interesting because they have implications for the evolution of cell fate specification mechanisms. For instance, germ cells are specified clonally and very early in sea urchins and the molecular mechanisms and cellular processes responsible are well studied. The absence of any cells with a similar transcriptional profile in seastar embryos implies no that early segregation of the germ line does not occur, and thus it is unlikely that localized maternal determinants are responsible. Given the wealth of information available about cell fate specification mechanisms in sea urchins, inferences about evolutionary differences could be made about other cell types as well. This seems to me a missed opportunity for the authors to highlight some interesting and informative aspects of their results.

We have added some new text into the “Essential Revisions #4”. We agree that timing changes are very important in our system and should be highlighted better in our text.

We have added some discussion about differences in the timing of germ line specification while also being careful to take into account the points made by Reviewer #1 about the lack of confidence in our identification in PGCs, which we have re-annotated as piwi3l+ Cells:

“We hypothesize that this could be caused by the differences in the timing of PGC formation. Previous work has shown that the genes typically used to identify them, such as piwi nanos, and pumillo, only begin to show cell-type-specific expression in P. miniata by the late gastrula stage when the posterior endocoelom forms (Fresques et al., 2014), a time point that lies outside our sampling window. Therefore, this cluster can be considered novel to S. purpuratus within our sampled window. “